# Gradient Routing: Masking Gradients to Localize Computation in Neural Networks

## Abstract

Neural networks are trained primarily based on their inputs and outputs, without regard for their internal mechanisms. These neglected mechanisms determine properties that are critical for safety, like (i) transparency; (ii) the absence of sensitive information or harmful capabilities; and (iii) reliable generalization of goals beyond the training distribution. To address this shortcoming, we introduce *gradient routing*, a training method that isolates capabilities to specific subregions of a neural network. Gradient routing applies data-dependent, weighted masks to gradients during backpropagation. These masks are supplied by the user in order to configure which parameters are updated by which data points. We show that gradient routing can be used to (1) learn representations which are partitioned in an interpretable way; (2) enable robust unlearning via ablation of a pre-specified network subregion; and (3) achieve scalable oversight of a reinforcement learner by localizing modules responsible for different behaviors. Throughout, we find that gradient routing localizes capabilities even when applied to a limited, ad-hoc subset of the data. We conclude that the approach holds promise for challenging, real-world applications where quality data are scarce.

## 1 Introduction

As AI systems become more powerful and more prevalent, there is an increasing need to explain and control the inner mechanisms governing their behavior. To address this challenge, some researchers aim to fully understand AI systems, either by reverse engineering the operations of conventionally trained models (Olah et al., 2020; Olsson et al., 2022) or with inherently interpretable architectures (Koh et al., 2020; Hewitt et al., 2023; Xin et al., 2022). This is not necessary. If we could control the mechanisms underlying a neural network's computation with respect to a limited set of safety-critical properties, such as hazardous information or the capacity for deception, that might be sufficient to make significant safety guarantees. Since manual specification of network internals is likely infeasible, there is a need for *mechanistic supervision*: the use of data to exert targeted influence over neural network internals.

To achieve mechanistic supervision, we propose gradient routing, a modification of backpropagation that uses data-dependent, weighted masks to control which network subregions are updated by which data points. By appropriately specifying these masks, a user can configure which parts of the network (parameters, activations, or modules) are updated by which data points (e.g. specific tokens, documents, or based on data labels). In this work, we apply gradient routing to a variety of problems:

**Section 4.1** We use gradient routing to split the encoding learned by an MNIST autoencoder into two halves, with each half representing different digits. We do the same for a CIFAR classifier in appendix B.1. In this way, we demonstrate supervised control of learned representations.

**Section 4.2** We apply gradient routing to localize features in language models. First, we train a model that can be steered by a single scalar value, showing that feature localization is possible, even with narrowly-scoped labels. Next, we present *Expand, Route, Ablate*, an application of gradient routing that enables robust removal of capabilities via ablation of a pre-specified network subregion. When data is partially labeled, the method outperforms all baselines, including data filtering, a gold standard of unlearning. Finally, we show that this unlearning method scales to a much larger (0.7B) model.

**Section 4.3** We apply gradient routing to the problem of scalable oversight (Amodei et al., 2016), where the aim is to train a performant policy despite limited access to reliable labels. We train a policy network by reinforcement learning to navigate to two kinds of grid squares in a toy environment, DIAMOND and GHOST. Using gradient routing, we localize modules responsible for these two behaviors. We show that we can steer the policy towards DIAMOND by ablating the GHOST module. Gradient routing trains steerable networks even when the amount of labeled training data is small (1%), and even when the policy is able to condition on the existence of labels. As a result, our method outperforms baselines based on behavioral supervision alone.

Throughout, we find evidence of an **absorption** effect, where gradient routing applied to narrow data localizes capabilities relevant to a broader superset of data. Absorption answers the question "if one has labels that are suitable for localizing undesirable computation, why not use those labels to filter the data?" When labels do not encompass all training data from which harmful capabilities arise (Zhu et al., 2009), filtering may be inadequate (Welbl et al., 2021), whereas absorption means that localization can still occur. Furthermore, localization influences model internals without modifying the loss function. This can enable scalable oversight when perfect supervision is not feasible.

We conclude by noting that black-box training techniques may be insufficient for high-stakes machine learning applications. Localization techniques, like gradient routing, may provide a solution.

## 2 BACKGROUND AND RELATED WORK

**Training to localize pre-specified capabilities.** Akin to gradient routing, work in modular machine learning trains modules to contain concepts or abilities determined in advance of training. Typically, modular architectures involve a routing function that selects modules to apply on a forward pass (Pfeiffer et al., 2023). Routing functions are often unsupervised, but some rely on metadata, inducing modules with known specializations (Waibel & II, 1992). For example, routing has been based on (i) the modality of data in multi-modal models (Pfeiffer et al., 2021), (ii) language (Pfeiffer et al., 2020; 2022; Fan et al., 2021), and (iii) low- vs. high-level control or task type in robotics (Heess et al., 2016; Devin et al., 2017). Gururangan et al. (2021) separate the training data of a language model by domain and assign one expert in each layer to a single domain. By disabling the expert for a domain, they are able to approximate a model that was not trained on the domain.

Other methods freeze the weights of a pre-trained model and train a new module, with the aim of localizing the task to the new module (Rebuffi et al., 2017; 2018; Houlsby et al., 2019; Bapna & Firat, 2019). Zhang et al. (2024) locate capabilities in models by learning a weight mask, transfer the identified sub-network to a randomly initialized model, then train as if from scratch. By choosing a suitable sub-network, they can, e.g., induce a vision model to identify ImageNet (Deng et al., 2009) classes by shape, not texture. Appendix J contains extended comparisons to select methods.

**Adversarial representation learning and concept erasure.** In order to control the information in learned representations, some have proposed to train feature extraction networks adversarially against discriminator networks that predict this information (Goodfellow et al., 2014; Schmidhuber, 1992; Ganin & Lempitsky, 2015; Ganin et al., 2016; Edwards & Storkey, 2015). Other methods attempt to remove concepts by modifying activations at inference time (Ravfogel et al., 2020; Belrose et al., 2023; Elazar et al., 2020; Bolukbasi et al., 2016). In contrast, gradient routing localizes capabilities during training, with the option of ablation afterward.

**Robust unlearning.** Machine unlearning seeks to remove undesired knowledge or abilities from a pre-trained neural network (Cao & Yang, 2015; Li et al., 2024). Typical unlearning methods are brittle in the sense that the unlearned abilities of the model can be recovered by fine-tuning on a tiny number of data points (Henderson et al., 2023; Sheshadri et al., 2024; Lynch et al., 2024; Liu et al., 2024; Shi et al., 2024; Patil et al., 2023; Lo et al., 2024; Lermen et al., 2023). Lee et al. (2024); Łucki et al. (2024) suggest that undesired concepts are more easily "bypassed" than thoroughly removed from model weights. In this paper, we pre-train models with gradient routing. Consequently, localized capabilities can be robustly removed via ablation. Tampering Attack Resistance (TAR) (Tamirisa et al., 2024) also targets robust unlearning in LLMs, but does so via fine-tuning.

Like gradient routing, some robust unlearning approaches prune or mask parts of the network most important for the target behavior. SISA (Bourtoule et al., 2021) trains multiple independent models based on a partition of the dataset and ensembles them at inference time. Similar to ablating a

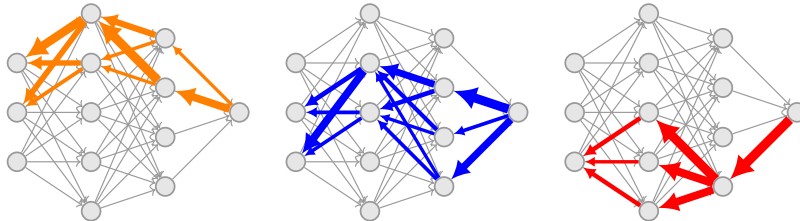

Figure 1: Gradient routing applies weighted masks to selectively block or re-weight gradients during backpropagation. By supplying different masks for different data, the user can induce specialization in network subregions. The figure shows three masks, which would correspond to three data points.

network subregion, a model can be dropped to achieve robust unlearning. Huang et al. (2024) and Pochinkov & Schoots (2024) remove neurons related to harmful behavior in order to restore the alignment of an adversarially fine-tuned language model. Guo et al. (2024) fine-tune the parameters of only the most important components for the task. Lizzo & Heck (2024) instead delete subspaces of the model parameters in order to remove specific knowledge. Unfortunately, Lo et al. (2024) find that models pruned to remove a concept can very quickly relearn the concept with further training. This may be because *identifying* the precise sub-network for a task post-hoc is very challenging, as evidenced by the modest success of "circuit discovery" in mechanistic interpretability thus far (Wang et al., 2023; Conmy et al., 2023; Miller et al., 2024; McGrath et al., 2023).

**Limits of data filtering for removal of undesired capabilities.** The challenge of limited or imperfect data labeling is ubiquitous in modern ML systems (Anwar et al., 2024). Obtaining comprehensive labels for harmful capabilities or behaviors is difficult. Current filtering approaches rely on simple heuristics and blacklists (Albalak et al., 2024). Automated toxicity filtering can inadvertently exclude valuable content from marginalized groups (Dodge et al., 2021; Chowdhery et al., 2023). Similarly, research on dataset filtering has shown that both rule-based approaches (Raffel et al., 2020) and narrow classifiers (Gehman et al., 2020; Solaiman & Dennison, 2021) struggle to effectively identify and filter harmful content (Welbl et al., 2021).

## 3 GRADIENT ROUTING CONTROLS WHAT IS LEARNED WHERE

**Gradient routing** applies data-dependent, weighted masks to gradients during backpropagation to configure **what** data (whether it be defined in terms of tokens, documents, or based on other labels) is learned **where** in the network (e.g. at the level of parameters, activations, or modules). The result is a model with a partially-understandable internal structure, where particular regions correspond to known capabilities. *Throughout this paper, we will use "route $X$ to $Y$" to mean "use gradient routing to limit learning updates for data points $X$ to region $Y$ of the neural network."*

Let $(\mathcal{V}, \mathcal{E})$ be the nodes and edges of the computational graph corresponding to a neural network and loss function, with $v(z)$ taken to be the output of node $v$ if $z$ is input to the network. Given a dataset $\mathcal{D} = \{z_i\}_{i=1}^n$, for each data point $z_i$, gradient routing requires the specification of a **gradient route** given by $\widetilde{\mathcal{E}}_i = \{\alpha_e^i \in \mathbb{R} : e \in \mathcal{E}\}$ and visualized in fig. 1. Define $\frac{\partial L(z)}{\partial v} \triangleq \frac{\partial L(\zeta)}{\partial v(\zeta)}|_{\zeta=z}$, the partial derivative of the loss $L$ with respect to the output of node $v$ when evaluated at input $z$. The routed derivative (denoted with a tilde) of the loss over a batch $\mathcal{B} \subseteq [n]$ is then defined recursively as $\frac{\widetilde{\partial} L(z_i)}{\widetilde{\partial} L} \triangleq 1$ for all $i \in \mathcal{B}$, and

$$\frac{\widetilde{\partial} L(z_i)}{\widetilde{\partial} v} \triangleq \sum_{u \in \text{child}(v)} \alpha_{(v,u)}^i \frac{\widetilde{\partial} L(z_i)}{\widetilde{\partial} u} \frac{\partial u(z_i)}{\partial v},$$

for all non-terminal nodes $v \in \mathcal{V} \setminus \{L\}$ and $i \in \mathcal{B}$. Choosing $\alpha_e^i \equiv 1$ recovers standard backpropagation. This weighting is only applied in the backward pass; the forward pass is left unchanged. Any gradient-based optimizer, like SGD or Adam (Kingma, 2014), can then be used to train with these modified gradients.

In practice, gradient routing masks need not be defined over every data point and edge in the computational graph. Instead, we limit masks to a small set of edges, like the outputs of specific MLP

neurons or the outputs of specific layers. Also, we typically assign gradient routes to data points based on membership in a coarse partition, like the forget set or retain set in an unlearning problem. Implementation is straightforward and efficient: algorithm 1 gives sample Pytorch (Paszke et al., 2019) code in which masking is applied to the outputs of sequential layers.

In all of our applications, masks are applied to activations of a few select layers. In most of our applications, these masks are binary, with 1's allowing the flow of gradients and 0's preventing the flow of gradients. Guidance for choosing these masks, and precise mask specifications for all our experiments, are given in appendix K. Informal descriptions are also given in the following section.

```python
def forward(self, x: Tensor, gradient_masks: list[Tensor]):
    for layer, mask in zip(self.layers, gradient_masks):
        act = layer(x)
        x = mask * act + (1 - mask) * act.detach()
    return x
```

Algorithm 1: Example of gradient routing implemented in PyTorch. For each batch of training data points x, a batch of `gradient_masks` corresponding to those data points is passed as well. The `detach()` method applies the stop-gradient operator, preventing gradients from being backpropagated through `act` but leaving its value unchanged.

## 4  APPLICATIONS

### 4.1  ROUTING GRADIENTS TO PARTITION MNIST REPRESENTATIONS

As a first example of feature localization via gradient routing, we train a simple MLP autoencoder on the MNIST handwritten digit dataset (LeCun et al., 1998) and use label-dependent stop-gradients to control where features for different digits are encoded. The goal is to obtain an autoencoder that reconstructs all digits (0–9) via an encoding that is made up of non-overlapping subcomponents corresponding to distinct subsets of digits. We choose subsets $\{0, 1, 2, 3, 4\}$ and $\{5, 6, 7, 8, 9\}$. To hint at the potential difficulty of this task, we note the encodings learned by an autoencoder trained on one of these sets admit low-error reconstructions on the other set, despite never being trained on it (details in appendix B).

We use a simple architecture of three-layer MLP modules with ReLU activations: an Encoder, a Decoder, and two "certificate" decoders. The Encoder processes a $28 \times 28$ image into a vector in $\mathbb{R}^{32}$, and the Decoder processes that vector into a $28 \times 28$ reconstruction. Each certificate is trained on *half* of the encoding, which takes values in $\mathbb{R}^{16}$. Certificate updates do not affect the encoding. If the Decoder can reconstruct a digit that a certificate cannot, this "certifies" that robust feature localization occurred (away from the half of the encoding the certificate was trained on).

We use gradient routing to train an encoding split such that the top half encodes digits 0–4 and the bottom half encodes digits 5–9. While training on all digits, we route digits 0–4 to the top half of the encoding and route digits 5–9 to the bottom half of the encoding. To induce specialization in the two halves of the encoding, we add the L1 norm of the encoding as a penalty term to the loss. The setup is shown in fig. 2a. The results, shown in fig. 2b and fig. 2c, are stark: while using the entire encoding allows the Decoder to reproduce all digits with low loss, the Certificate is only able to reproduce 5–9 from the bottom half of the encoding, as desired. Furthermore, the certificate's learned predictions for digits 0–4 are approximately constant. This suggests that we have successfully eliminated most information relevant to digits 0–4 from the encoding. Appendix B contains experiment details, ablations, and an extension to a ResNet (He et al., 2016) trained for CIFAR image classification (Krizhevsky et al., 2009).

### 4.2  LOCALIZING TARGETED CAPABILITIES IN LANGUAGE MODELS

In this section, we show that gradient routing applied to a small set of tokens can be used to localize broader features or capabilities in Transformer (Vaswani, 2017) language models. This is first

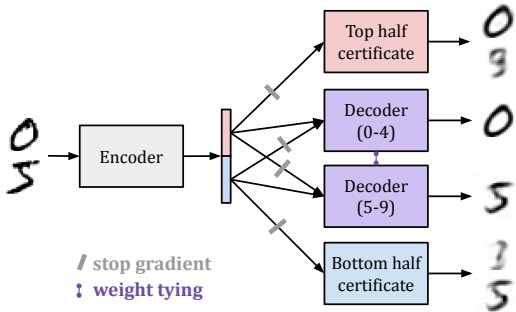
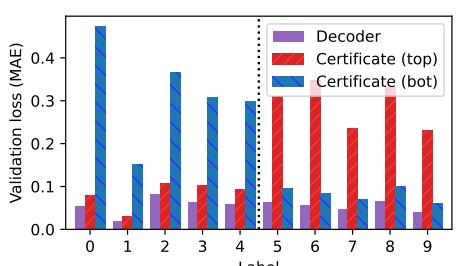

(a) An autoencoder trained to encode digits 0–4 in the top half encoding and digits 5–9 in the bottom half. The full encoding is processed by a single Decoder module trained with gradient routing; we illustrate this using weight tying and stop gradients. The two certificates are trained to reconstruct all digits from different halves of the encoding.

(b) Average (across 20 runs) validation set reconstruction losses, measured as the pixel-wise mean absolute error (MAE) for the Decoder and the certificates, demonstrating successful localization of information about digits. Run-to-run variation is negligible.

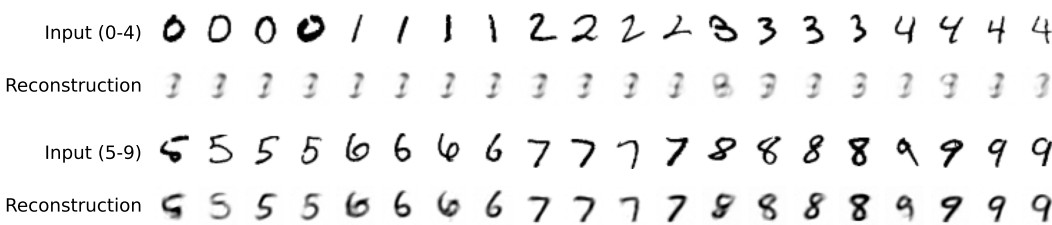

(c) Bottom half certificate reconstructions from the validation set. The near-constant prediction of the certificate on digits 0–4 illustrates the absence of information about those digits from the bottom half of the encoding. Top half reconstructions are given in fig. 6 in the appendix.

Figure 2: Gradient routing induces a clean split in the encodings of a simple MLP autoencoder trained on MNIST digits. By applying data-dependent stop-gradients and L1 regularization, the top half of the encoding comes to represent digits 0–4 only, and the bottom half of the encoding comes to represent digits 5–9 only.

demonstrated in terms of model activations, then applied to MLP layers for the purpose of robust unlearning.

### 4.2.1 STEERING SCALAR: LOCALIZING CONCEPTS TO RESIDUAL STREAM DIMENSIONS

Elhage et al. (2021) frames the inter-block activations of a Transformer, or *the residual stream*, as the central communication channel of a Transformer, with all layers "reading from" and "writing into" it. Usually, the standard basis of the residual stream is indecipherable, with the axes not corresponding to interpretable concepts. We pre-train a 20-layer, 303M parameter Transformer on the FineWeb-Edu dataset (Penedo et al., 2024) while routing the gradients for all _California[1] tokens to the $0^{th}$ entry of the residual stream on layers 6–18. On token positions predicting _California, we mask gradients (to zero) on every residual stream dimension except the $0^{th}$ in layers 6–18. This masking causes the learning updates for those token positions to be localized to the weights that write into the $0^{th}$ dimension of the residual stream. After training, we look at which tokens' unembedding vectors have the highest cosine similarity with the one hot vector for the $0^{th}$ entry of the residual stream. We find that _California has the highest cosine similarity, followed by California, _Californ, _Oregon, _Colorado, _Texas, _Florida, _Arizona, _Sacramento, and _Los; see appendix D for the top 300. These tokens all have semantic similarity to California, but gradient routing was not applied to them. This shows that gradient routing localizes broader semantic concepts, rather than the narrow set of explicitly-routed tokens.

Past work on activation steering (Turner et al., 2023; Rimsky et al., 2024) computed (non-axis aligned) *steering vectors* specified by $d_{model}$ different values. However, since we localized

---

[1]We use a leading _ to represent a leading space before a token.

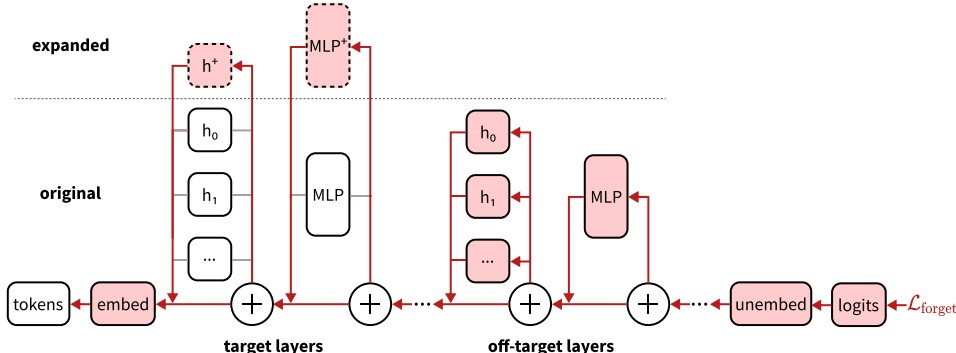

Figure 3: Backpropagation in the Route step of Expand-Route-Ablate, showing the flow of gradients through a Transformer for tokens in the forget set. This assumes a learning rate of zero for the original dimensions in target layers. Gradients for retain tokens are unmodified. Additional dimensions, shown with dashed outlines, were added to **target** layers in the MLP and attention blocks, and will be removed after training in the Ablate step. All modules participate in the forward pass.

California-related concepts to the $0^{th}$ dimension of the residual stream, we can steer the model to generate text related to California by adding a single scalar value to the $0^{th}$ entry of the residual stream during the forward pass. Appendix D provides steered model completions.

### 4.2.2 GRADIENT ROUTING ENABLES ROBUST UNLEARNING VIA ABLATION

Robust unlearning (Sheshadri et al., 2024) means training models that lack the internal mechanisms or "knowledge" required for certain tasks, as opposed to merely performing poorly on those tasks. To address this open problem, we show that gradient routing can be used to localize capabilities to a known region of the network. Then, that region can be deleted to remove those capabilities. We find that gradient routing excels in situations where data is only partially labeled.

To enable comprehensive comparisons, our initial study on robust unlearning applies gradient routing to a small (28M parameter) Transformer. This model is trained on an LLM-generated dataset of simple children's stories based on the TinyStories dataset (Eldan & Li, 2023; Janiak et al., 2024). We partition the data into a **forget set** made up of any story containing one of the keywords "forest(s)", "tree(s)", or "woodland(s)", and a **retain set** made up of all other stories; the forget set constitutes 20% of the training data. An example story is given in appendix C. The goal is to train a model that performs well on the retain set but poorly on the forget set, and whose forget set performance is not easily recoverable by fine-tuning.

To do this, we route specific forget tokens to designated MLP neurons using a three-step process termed Expand, Route, Ablate (ERA): **1. Expand:** Increase the dimensionality of the model by adding randomly-initialized neurons to particular *target layers*. **2. Route:** train the model from scratch by supervised learning on next-token prediction. On select tokens in forget stories, reduce the learning rate (possibly below 0) in the original dimensions of the model at the target layers. Figure 3 illustrates the routing step. **3. Ablate:** delete the additional neurons. Post-ablation, apply a very small number of steps of fine-tuning on retain data to correct for degradation caused by ablation.

**Experiments.** We compare ERA against three unlearning methods. (a) *Data filtering* discards a model trained on all data, then re-trains from scratch on retain data only. By not training on forget data, it serves as a gold standard for unlearning. (b) *Representation misdirection for unlearning* (RMU) (Li et al., 2024) fine-tunes a model trained on all data to corrupt its internal representations of forget data. It is a conventional post-hoc unlearning method. (c) *DEMix plus ablation* replaces all MLPs with domain expert mixture layers (Gururangan et al., 2021) comprised of an MLP that operates only on retain data and an MLP that only operates on forget data; after training the whole model on all data, the forget expert is ablated. DEMix plus ablation serves as an alternative localization-based approach.

Models are trained with different proportions of forget data labeling to simulate the challenges of real-world data labeling (Anwar et al., 2024). When a forget sample (a story) is not labeled, it is

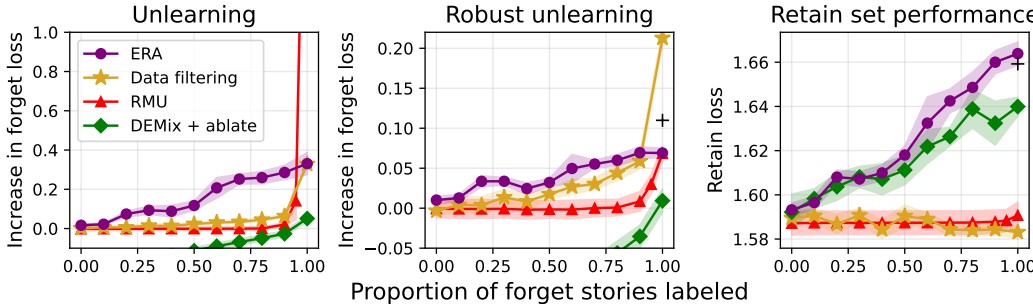

Figure 4: Effect of unlearning methods on forget and retain validation loss depending on the proportion of forget samples labeled. Highlighted regions denote 95% C.I. for the mean across at least $N = 5$ training runs. *Left*: how much each method increases forget loss after it is applied. For ERA and DEMiX + ablate, this is pre- vs. post-ablation. *Center*: how much forget loss increases after a method is applied and the model is fine-tuned on 64 forget stories. (The minimum validation forget loss over fine-tuning is reported.) *Right*: the retain set performance after applying each method. *Note: we include an additional data point for RMU at 0.95 of forget stories labeled.*

treated as a retain sample for training and unlearning purposes. Validation data is always labeled correctly. We report three metrics: *unlearning* is the difference in forget loss before and after unlearning is applied; *robust unlearning* is the difference in forget loss before unlearning is applied and after it is applied *and* the model is retrained on 64 forget samples; *retain set performance* is the loss on the retain set after applying the method.

**Results.** When labeling is limited (<100%), ERA dominates, outperforming even the gold-standard data filtering baseline (fig. 4, *left*), both in terms of unlearning and robust unlearning. This comes at the cost of degraded retain set performance, proportional to the amount of data that routing was applied to (fig. 4, *right*). DEMiX + ablate, the localization-based competitor, has negative unlearning in all settings except 100% labeling. This is because the forget expert is trained only on labeled forget stories, whereas the retain expert trains on the much-larger retain set and unlabeled forget stories.

At 100% oversight, the top performers are as expected: RMU, a conventional unlearning method, attains the highest loss after unlearning but before being retrained on forget data. Data filtering, a gold standard, is the most robust to retraining. In contrast, most of RMU's unlearning is undone by retraining. Although ERA achieves higher retrained forget loss than RMU (appendix C.1, fig. 9), when correcting for the general performance degradation of ERA, ERA robust unlearning matches that of RMU (fig. 4, *center*). However, by combining ERA and RMU (indicated by a "+"), we achieve better robust unlearning than either method alone. Further discussion, experiment details, hyperparameters, and results are given in appendix C.

### 4.2.3 SCALING ROBUST UNLEARNING TO LARGER LANGUAGE MODELS

Gradient routing can localize capabilities in larger models. Motivated by the dual-use nature of AI (Urbina et al., 2022), we would like to train useful models that lack certain harmful capabilities. Here, we seek to localize and remove bioweapon-related capabilities in a 0.7B parameter Transformer. To do this, we route 20 tokens related to virology[2] to the 0th through 79th MLP dimensions on layers 0 through 7 of the Transformer. Appendix E provides further details.

Table 1 evaluates the model on a validation split of regular FineWeb-Edu data and on some of the WMDP-bio (Li et al., 2024) forget set. Ablating the target region of the network increases loss greatly on both datasets. We then fine-tune the model on a train split of FineWeb-Edu for 32 steps to restore some performance. Finally, we retrain for twenty steps on a separate split of two WMDP-bio forget set datapoints, as in Sheshadri et al. (2024), and report the lowest loss on the validation split of the WMDP-bio forget set.

---

[2]Specifically, we route on `COVID`, `_COVID`, `RNA`, `_infections`, `DNA`, `_genome`, `_virus`, `_gene`, `_viruses`, `_mutations`, `_antibodies`, `_influenza`, `_bacteria`, `PCR`, `_cell`, `_herpes`, `_bacterial`, `_pathogens`, `_tumor`, and `_vaccine`.

Table 1: Performance of a language model trained with gradient routing on virology tokens. The final column evaluates the model after fine-tuning on FineWeb-Edu and then retraining on two examples from the WMDP-bio forget set, choosing the retraining step with the lowest loss. The increase in loss on (the validation split of) the WMDP-bio forget set is much higher than the increase in loss on FineWeb-Edu data, demonstrating successful localization and robust unlearning. Intriguingly, this increase persists even when excluding routed tokens from the loss calculation, showing a broader localizing effect.

| Dataset | Loss | Ablated loss ($\Delta$) | Retrained loss ($\Delta$) |
|---|---|---|---|
| WMDP-bio forget set ↑ | 2.596 | 4.283 (+1.687) | 2.778 (+0.182) |
| WMDP-bio forget set (sans routed toks)↑ | 2.567 | 4.205 (+1.638) | 2.738 (+0.171) |
| FineWeb-Edu ↓ | 2.925 | 4.864 (+1.939) | 2.957 (+0.032) |

The results are striking: even after retraining on virology data, loss increases much more on the WMDP-bio forget set (+0.182) than on FineWeb-Edu (+0.032), demonstrating successful localization and robust removal of virology capabilities. A natural concern would be that ablation merely decreased probabilities on the routed tokens, without decreasing overall virology capabilities. To test this, we measured cross-entropy loss on the forget set excluding the 20 tokens we routed on. Even after this exclusion, the loss increase is still much higher than the increase on FineWeb-Edu (+0.171 vs. +0.032). This shows that gradient routing generalizes beyond limited labels.

### 4.3 LOCALIZING BEHAVIORAL MODULES ENABLES SCALABLE OVERSIGHT IN REINFORCEMENT LEARNING

In complex settings, reliable data labels are often scarce, especially when labeling requires human input (Zhu et al., 2009; Stiennon et al., 2020; Bai et al., 2022; Baker et al., 2022). The aim of *scalable oversight* (Amodei et al., 2016; Finn et al., 2016) methods is to leverage limited supervision in order to effectively learn from unlabeled data. A challenge to scalable oversight is that label quality or availability may be correlated with task difficulty, as in *weak-to-strong generalization* (Burns et al., 2024). The correlation poses a key challenge to scalable oversight of a reinforcement learner: the learner may indirectly condition its behavior on the extent to which we can oversee it, exploiting limitations of the labels (Krakovna et al., 2020; Karwowski et al., 2024). In this section, we show that gradient routing can be used to achieve scalable oversight even in this challenging setting: by routing labeled data to submodules, we leverage unlabeled data to obtain a performant policy that does not exploit the limits of our labels.

We use gradient routing to train a policy to reach two types of squares in a gridworld, DIAMOND and GHOST. The aim is to produce a policy that reliably navigates to DIAMOND and not GHOST. However, access to the type of square reached, or *oversight*, is available only for some episodes. Furthermore, we allow the policy to condition on the presence or absence of oversight, modeling an agent that observes features that are not accessible to the supervisor. Because of this information asymmetry, the desired agent behavior cannot be specified by *any* reward function. Despite this challenging setup, gradient routing induces efficient learning by causing learning updates from the two behaviors to accumulate in different parts of the network.

**Partial oversight environment.** The environment is based on a $5 \times 5$ gridworld Markov Decision Process (Puterman, 1990) with two randomly placed terminal squares: DIAMOND and GHOST. At each step, the agent can move north, west, south, or east. Episodes end upon reaching a terminal square or after 32 steps. The environment state includes a boolean variable for every grid square that indicates terminal squares that are under oversight. These indicators are sampled randomly and independently at the beginning of each episode according to the *oversight level* $p \in [0, 1]$. If an episode ends with the agent reaching a terminal grid square with the oversight indicator set to FALSE, then the reward function does not have access to the square type labels (DIAMOND, GHOST) for that entire episode. The policy takes the entire state as input, including oversight indicators.

**Architecture and training.** The policy network is a Mixture of Experts (MoE) network Eigen et al. (2013) with two MLP experts (a DIAMOND expert and a GHOST expert) and a MLP gating network, each of which takes the environment state as input. The expert outputs are combined via a convex

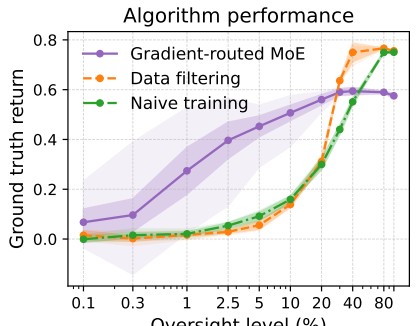
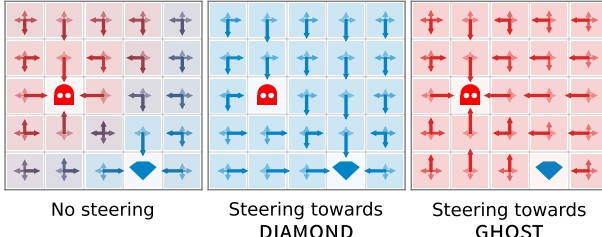

(a) Average stepwise ground truth returns at different oversight levels, evaluated at the end of training. (Highlights: 95% C.I. for the mean across runs, 5th/95th quantiles.)

(b) The gradient-routed MoE policy in an instance of the environment. Neither terminal square is under oversight. Nevertheless, steering induces desired behavior. Arrows: N/W/S/E action probabilities. The policy was trained under 10% oversight.

Figure 5: Using gradient routing and a mixture of experts layer, we train an agent that can be steered towards desirable behaviors, even when oversight is limited. Policies were evaluated on 512 randomly generated environments, each run for 32 steps.

combination with weight computed by the gating network. This value is processed by a linear layer to produce policy logits. Appendix F has details.

The policy network is trained with REINFORCE (Williams, 1992). When a terminal square with oversight is reached, we mask gradients so that the learning update is restricted to either the DIAMOND expert or GHOST expert depending on which terminal square the agent reached; we also train the gating network to activate only the module through which the gradients were routed. In episodes without oversight, we perform only a normal REINFORCE update with regular backpropagation.

**Experiment setup.** We compare gradient routing with two baselines trained with behavioral supervision. *Data filtering* trains only on episodes where the end state is observed, discarding unlabeled episodes. *Naive training* utilizes all episodes, using a reward of −1 when reaching GHOST under oversight, and +1 when reaching a terminal square in any other case. Naive training can be understood in terms of process supervision (Uesato et al., 2022): it rewards outcomes (finishing the episode quickly) and evaluates process (which terminal state reached) only when possible. We compare the methods by the average stepwise *ground truth return* they attain; the ground truth return is 1 for reaching DIAMOND, −1 for reaching GHOST, and 0 otherwise, with a discount factor of 0.97 to reward shorter paths. Policies are trained for 20,000 algorithm steps (328M environment steps).

**Results.** Gradient routing successfully localizes expert modules responsible for navigating to DIAMOND and GHOST (fig. 5b), even when evaluated at states where oversight is not present. Utilizing the DIAMOND expert outperforms baselines and achieves nontrivial performance at oversight as low as 1% (fig. 5a). At 5% oversight, the DIAMOND expert reaches performance comparable to that of the other methods trained with 6x greater oversight. Importantly, we note that improved baseline performance at high oversight levels is largely a consequence of a greater proportion of evaluation environments having oversight, rather than good baseline performance when oversight is lacking.

Further discussion, learning curves at 10% oversight, experiment details, and ablations are given in appendix F. We find that naive training exploits the limits of oversight, learning to avoid GHOST with oversight and seek nearby GHOST without oversight. We also find that data filtering fails to learn *even in the limit of train time*. In contrast, an *oracle filtering* baseline, which is able to observe all end states but downsamples the training data based on the oversight level, eventually achieves convergence. In summary, gradient routing is strictly better than feasible baselines at low oversight.

## 5 DISCUSSION

**Gradient routing induces absorption.** Routing a subset of the data related to some knowledge or capability appears to localize that knowledge or capability more generally. This held for an i.i.d.

subset of the data (TinyStories unlearning in section 4.2.2), and for semantically limited data (steering scalar in section 4.2.1, virology unlearning in section 4.2.3, scalable oversight in section 4.3). Notably, this effect did not hold for DEMix, a modularity method in which localized modules are sequestered so that only one (per layer) participates in each forward pass. To explain these observations, we posit *absorption*: (i) routing limited data to a region creates units of computation or features that are relevant to a broader task; (ii) these units then participate in the model's predictions on related, non-routed data, reducing prediction errors on these data, so that (iii) the features are not learned elsewhere. Absorption may also amplify the features causing it. When data labels are semantically or quantitatively limited, absorption means that gradient routing can be useful even in cases where conventional training or data filtering methods are inadequate.

**Mechanistic supervision avoids Goodharting.** When the ability to label (or score) outcomes is imperfect, attempting to suppress undesirable behavior via behavioral training is fraught (Goodhart, 1984; Karwowski et al., 2024). In contrast, gradient routing provides mechanistic supervision, influencing training without modifying the behavioral objective. We showed this empirically in section 4.3, where an agent trained naively based on partially observed outcomes learned to pursue the user-desired outcome when observed but not otherwise. On the other hand, gradient routing utilized the same observations to induce the desired behavior mechanistically.

**Entangled capabilities motivate gradient routing.** In many machine learning problems, capabilities are *entangled* in the sense that there are connections or dependencies between the computation learned to perform different tasks (Arora & Goyal, 2023; de Chiusole & Stefanutti, 2013). Entanglement might occur because certain capabilities or behaviors are reinforced by a broad range of training objectives (Omohundro, 2008; Turner et al., 2021; Krakovna et al., 2020). More simply, capabilities required to perform desired tasks may overlap with those required to perform undesired tasks. For example, biological knowledge entails much of the knowledge required to construct biological weapons. For this reason, filtering or training against bioweapon-specific data might not prevent a network from learning enough to create bioweapons from general biology sources or would require such broad filtering so as to render the model useless at biology in general. In principle, gradient routing can avoid this by localizing a more limited subset of capabilities, then ablating them.[3] Alternatively, gradient routing could be employed to robustly detect when a given capability is being used by the model (when a localized module strongly activates). This kind of monitoring would provide an avenue for the application of access controls (Sandhu & Samarati, 1994; Samarati & de Vimercati, 2001) to high-stakes AI deployment, as explored in appendix L.

**Limitations and future work.** (a) Gradient routing's performance is sensitive to its hyperparameters: what data to route on, what regions to localize to, and what mask weights to use. This makes it hard to balance retain set performance vs. unlearning, for example. We suspect that methodological improvements will reduce this sensitivity. (b) In our experiments with language models, we route gradients on a token-by-token basis, ignoring neighboring tokens. This naive strategy is surprisingly effective. However, it is plausible that contextual information will be critical in some problems, necessitating routing strategies that depend on entire sequences. Finding practical ways of choosing what data to route in order to localize broad capabilities is an intriguing open problem. (c) Our empirical results for scalable oversight pertain to a simplistic, narrow setting. Furthermore, our method for scalable oversight requires that the ablated policy produce coherent behavior. This does not hold in general, so scaling oversight via localization may require new ideas. (d) We elaborate on application-specific limitations in appendix A.

## 6  CONCLUSION

Gradient routing enables data-driven supervision of the internal mechanisms learned by neural networks. Even when this supervision is based on simple or limited data labels, it can achieve robust unlearning of pre-specified capabilities and scalable oversight. Consequently, gradient routing may facilitate the safe deployment of AI systems, particularly in high-stakes scenarios where black-box methods are insufficiently robust.

---

[3]Entangled capabilities present fundamental tradeoffs: the removal or attenuation of a capability may *necessarily* harm capabilities entangled with it. The claim is not that gradient routing avoids this tradeoff, but that it plausibly enables more efficient tradeoffs.

ACKNOWLEDGMENTS

Anonymized for review.

REPRODUCIBILITY STATEMENT

We include detailed descriptions of experiment settings in the appendix. Anonymized code to reproduce our results is presented as-is at:

https://anonymous.4open.science/r/factored-representations-3035/README.md.

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

# APPENDIX TO GRADIENT ROUTING: MASKING GRADIENTS TO LOCALIZE COMPUTATION IN NEURAL NETWORKS

## A EXTENDED DISCUSSION OF APPLICATION-SPECIFIC LIMITATIONS AND FUTURE WORK

**MNIST autoencoders.** The cleanly separated MNIST autoencoder representations depicted in fig. 2c depend on the problem setup (e.g. the choice to *not* use data augmentation, like rotations) and use of heavy L1 regularization on the encoding vector. L1 regularization is required because, by default, a regular MLP autoencoder trained on a subset of MNIST digits retains information necessary to decode other digits.

For a wide range of hyperparameters, we find that gradient routing achieves *quantitative* representation splitting: the Certicate's reconstruction of digits 0–4 has higher average loss than its reconstructions of digits 5–9 for a wide range of settings, including different partitions of the digits. However, outside the specific hyperparameters chosen for the results in the main body of the paper, the *qualitative* results are poorer: the visual difference in reconstruction quality between the different digit subsets is less stark than in fig. 2c. We take this to highlight the problem-dependent characteristics of feature localization. In the case of autoencoding handwritten digits, separation of features for encoding different digits is "unnatural," so achieving it requires a specific setup and heavy regularization.

**Language models.** We speculate that gradient routing on particular tokens introduces an "internal tug of war" between the expanded and original dimensions of the model (these dimensions depicted in fig. 3), where parameter updates in the original dimensions consistently decrease the logits for routed tokens and parameter updates in the expanded dimensions increase logits for routed tokens. This effect can be understood as a consequence of the mismatch between the implicit estimands (learning targets) for the original and expanded dimensions. We were concerned that this effect, rather than localization of capabilities, explained the post-ablation increase in forget loss. However, preliminary measurements suggest that this is not the case. For example, we find that the loss of ERA models is higher on average on *non-routed* forget tokens than a pure model, whereas it is lower on average on *routed* tokens. In general, the learning dynamics of gradient routing remain an open question.

If routing one token to a dimension of the residual stream creates an interpretable, axis-aligned feature as discussed in section 4.2.1, then routing many tokens to many neurons could produce a neural network with transparent internal representations. These representations might be made up of "individual neurons. . . [that] corresponded to cleanly interpretable features of the input," as imagined in Elhage et al. (2022), or they could be organized in different ways. In principle, gradient routing provides a straightforward means of achieving this. However, we suspect that naive attempts to localize large numbers of concepts to unique regions will lead to high training loss.

**Scalable oversight.** Our reinforcement learning results demonstrate the promise of a localization-based strategy for scalable oversight, but further empirical and conceptual work is needed. The toy environment we use is simple, lacking the complexity and asymmetries of real-world problems. Additionally, our proposed solution relies on the fact that ablating an otherwise-active module of a policy network produces a policy with coherent behavior, which may not be true in practice (and isn't true in general, in principle). We discuss these considerations in appendix G.

## B MNIST AUTOENCODER DETAILS AND ABLATIONS

**Model architecture.** The Encoder, Decoder, and certificates are all three-layer MLPs. The layer sizes for the Encoder produce data with shapes ($28 \times 28$, 2048, 512, 32) and for the decoder, data with shapes (32, 512, 2048, $28 \times 28$). All hidden layers use ReLU activations. The final layer of the Encoder is linear. The final layer of the decoders is affine.

**Training.** The model was trained for 200 epochs on the 60,000 image training part of the MNIST dataset (LeCun et al., 1998) with batch size 2048. Images were normalized to have mean and standard deviation 0.5. No data augmentation was used. Optimization was performed with Adam

Figure 6: The *top half* certificate reconstructions corresponding to fig. 2a, showing that the top half of the encoding contains information necessary to accurately reconstruct digits 0–4 while containing practically no information relevant to reconstructing digits 5–9.

(Kingma, 2014) with learning rate 1e-3, $\beta = (0.9, 0.999)$, and weight decay 5e-5. All modules are initialized with the default Pytorch initialization.

The loss used was pixel-wise mean absolute error, with a penalty term for the L1 norm of the encoding and a penalty term for the sum of absolute correlations (across batch elements) between the top and bottom half of the encoding. For a batch of data indexed $i = 1, \ldots, n$ and encoding size 32, denote data points by $x_i$, encodings as $\widehat{z}_i$, and Decoder outputs as $\widehat{x}_i$. Then for $\lambda = 0.003$ and $\gamma = 0.1$, the loss used to train the autoencoder is $\mathcal{L} = \mathcal{L}_{\text{reconstruction}} + \lambda \cdot \mathcal{L}_{\text{L1}} + \gamma \cdot \mathcal{L}_{\text{Correlation}}$, where

$$\mathcal{L}_{\text{reconstruction}} = \frac{1}{28^2 \cdot n} \sum_{i=1}^{n} \|x_i - \widehat{x}_i\|_1,$$

$$\mathcal{L}_{\text{L1}} = \frac{1}{n} \sum_{i=1}^{n} \|\widehat{z}_i\|_1, \text{ and}$$

$$\mathcal{L}_{\text{Correlation}} = \frac{1}{16^2} \sum_{k=1}^{16} \sum_{h=17}^{32} \frac{\sum_{i=1}^{n} |\widehat{z}_{i,k} - \overline{z}_{\star,k}| |\widehat{z}_{i,h} - \overline{z}_{\star,h}|}{\sqrt{\sum_{i=1}^{n} (\widehat{z}_{j,k} - \overline{z}_{\star,k})^2} \sqrt{\sum_{i=1}^{n} (\widehat{z}_{j,h} - \overline{z}_{\star,h})^2}},$$

with $\overline{z}_{\star,k} = n^{-1} \sum_{i=1}^{n} \widehat{z}_{i,k}$. *Note: this equation does not include gradient routing, which is an intervention applied to gradients when backpropagating $\mathcal{L}_{\text{reconstruction}}$ through $\widehat{z}_i$.*

**Additional results and ablations.** Additional findings are given below. Many of them reference table 2, which provides results from ablation experiments.

- For a given set of hyperparameters, the run-to-run variability induced by random neural net initialization and data shuffling is small. For our main results (setting 1 in table 2), the 5th and 95th quantiles (across runs) of the average (over digits) final validation loss are (0.31, 0.33) for digits 0–4 and (0.08, 0.09) for 5–9.
- We find that training a regular autoencoder on a subset of digits, without regularization or gradient routing, results in an encoding that admits reconstructions of the digits that were not trained on (setting 8 of table 2).
- Inclusion of the correlation penalty helps split representations but is not necessary (compare setting 1 and setting 3 of table 2). However, regularization is necessary to achieve splitting (compare settings 1 and 2 to settings 4 and 5 of table 2).
- We find that we can learn separate "split" encodings of MNIST digits simply by training autoencoders on subsets of digits with a high L1 penalty, rather than applying gradient routing (setting 7 of table 2). However, gradient routing is still able to produce split encodings even in a more challenging setting where only one of the subsets of digits is routed, while the other has its gradients flow through the whole encoding (setting 6 of table 2, shown in fig. 7 and fig. 7c).
- (Not presented in this document) For most digit partitions that we tried (other than 0–4 and 5–9), we were able to reproduce results similar to those given in fig. 2 without modifying hyperparameters. Generally, the results were quantitatively comparable to, but less visually striking than, those shown in fig. 2c. We were even able to split the encoding into 10 parts, one per digit.

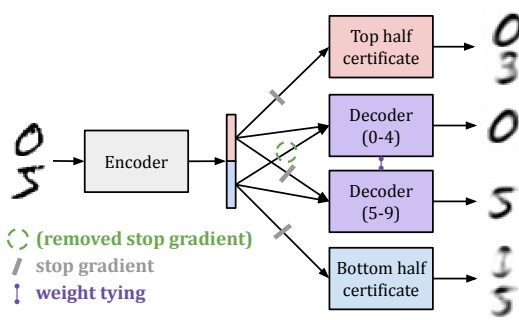 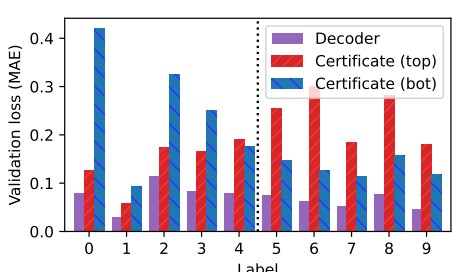

(a) A variant of an autoencoder trained to encode digits 0–4 in the top half encoding and digits 5–9 in the bottom half. Unlike the original training setup (fig. 2a), this variant only routes gradients for digits 5–9.

(b) Validation set reconstruction losses, measured as the pixel-wise mean absolute error (MAE) for the Decoder and the certificates.

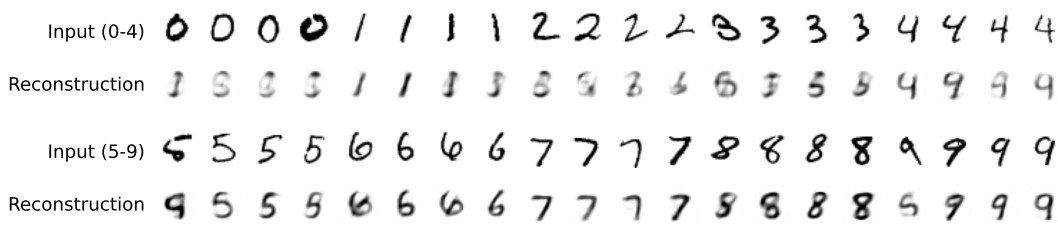

(c) Bottom half certificate reconstructions from the validation set.

Figure 7: A variant of the MNIST gradient routing experiment from section 4.1. In this version, gradients from all digits (rather than merely 5–9) are allowed to flow through the bottom half of the encoding. Since the goal is to isolate the representations for digits 0–4 to the top half encoding, the inclusion of digits 0–4 in learning updates for the bottom half encoding makes the problem more challenging. However, by increasing the strength of the L1 penalty applied to the bottom half encoding, we still achieve splitting.

Table 2: The average (over 20 runs) reconstruction losses for the bottom half certificate for different MNIST autoencoder training settings. Approximate 95% confidence intervals are given in parentheses. Default regularization settings are an L1 penalty on the encoding with weight 3e-3, and a penalty on the sum of absolute correlations between the top and bottom half entries with weight 0.1. Gradient routing (Setting 1) is presented in the main body of the paper and uses the default regularization. Settings marked with "separate Decoders" trained a Decoder on digits 0–4 and a different Decoder on digits 5–9 (equivalent to removing weight tying in fig. 2a). Setting 6 is the same as Setting 1, with two modifications: no stop gradients are used on the bottom half encoding, and the L1 penalty is increased to 2e-2 on the bottom half encoding. Setting 6 is depicted in fig. 7.

| Setting | Loss: 0–4 | Loss: 5–9 |
|---|---|---|
| 1. Gradient routing | 0.32 (±0.02) | 0.08 (±0.00) |
| 2. Gradient routing, separate Decoders | 0.33 (±0.02) | 0.07 (±0.00) |
| 3. Gradient routing, no correlation penalty | 0.28 (±0.02) | 0.11 (±0.01) |
| 4. Gradient routing, no regularization | 0.32 (±0.02) | 0.32 (±0.01) |
| 5. Gradient routing, no regularization, separate Decoders | 0.09 (±0.01) | 0.08 (±0.00) |
| 6. Gradient routing, bottom half encoding trained on 0–9 | 0.23 (±0.02) | 0.13 (±0.01) |
| 7. No gradient routing, L1 penalty 1e-3, trained on 5–9 only | 0.27 (±0.02) | 0.11 (±0.00) |
| 8. No gradient routing, no regularization, trained on 5–9 only | 0.08 (±0.01) | 0.08 (±0.00) |
| 9. No gradient routing, with regularization | 0.13 (±0.01) | 0.13 (±0.01) |
| 10. No gradient routing, no regularization | 0.08 (±0.01) | 0.09 (±0.00) |

### B.1 EXTENDING MNIST EXPERIMENTS TO CIFAR100 CLASSIFICATION

Can gradient routing be used to split representations more generally, or is MNIST a special case? To answer this question, we run the same experiment with a different model, dataset, and task.

**Experiment setup.** We train a ResNet (He et al., 2016) on the CIFAR100 (Krizhevsky et al., 2009) dataset to classify images, and apply gradient routing based on class label (in this case, whether the label is in 0–49 or 50–99). Using the original 34-layer ResNet architecture, we designate the convolutional layers as the Encoder, and the remaining pooling and linear layer as the Decoder (in this case, the Decoder is a classifier over 100 image classes, such as *otter*, *castle*, *oak*, *train*, etc.). We add two certificates, which are of the same type as the Decoder, except with the number of input channels halved. The Decoder, Encoder, and certificates are trained as shown in fig. 2a, with the encoding partitioned into halves along the channel dimension. As with MNIST, we include a penalty term in the loss that is the weighted L1 norm of the encoding. We also compare with setup that is identical, except gradient routing is not performed and no L1 penalty is applied.

**Results.** The results are given in fig. 8. We see a stark localizing effect of gradient routing and L1 regularization, as well as a significant reduction in validation accuracy. Cursory ablations (not shown) suggest that both localization and the performance hit are due to gradient routing, not the use of L1 penalty. The L1 penalty simply enhances gradient routing's ability to localize features. This is consistent with the findings from the extensive MNIST ablations given in appendix B, table 2.

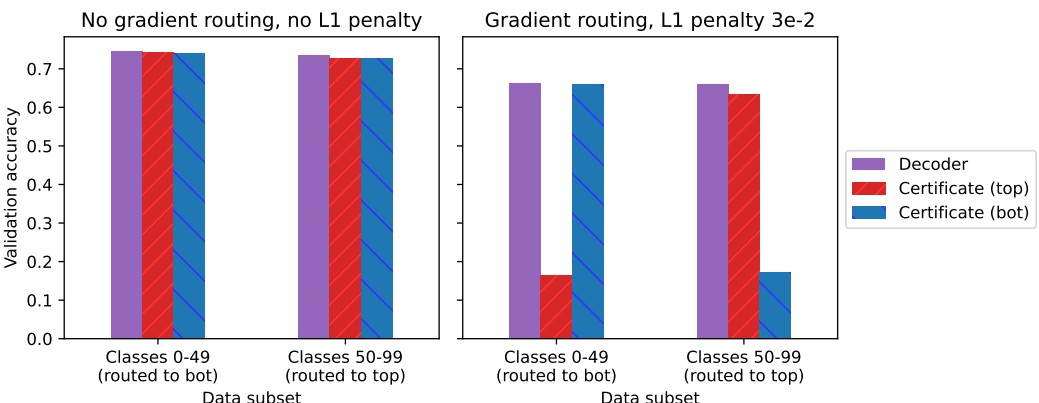

Figure 8: Average validation set performance for different ResNet classifiers: the Decoder, trained on all channels of the encoding, and the top and bot certificates, trained on their respective halves of the channels of the encoding. Variability in these estimates is small in contrast to the differences between metrics (for each of the gradient routing metrics, 95% confidence interval widths based on $N = 4$ runs are between 0.03 and 0.07).

**Discussion.** Our results show that in a different domain, the same gradient routing strategy achieves the same kind of outcome, with similar dynamics to the MNIST case. Interestingly, we also found that localization at middle layers works, but requires the addition of a single convolutional layer at the beginning of the decoders to break the residual connection.

**Details.** Our ResNet implementation is adapted from `https://github.com/kuangliu/pytorch-cifar/blob/49b7aa97b0c12fe0d4054e670403a16b6b834ddd/models/resnet.py`. The model was trained for 200 epochs on the 50,000 image training split of the CIFAR100 dataset (Krizhevsky et al., 2009) with batch size 128. The following random augmentations were applied during training: random cropping, horizontal flipping, and image normalization. Optimization was performed by SGD with learning rate 0.1, momentum 0.9, and weight decay of 5e-4. The learning rate was decayed according to cosine learning rate annealing over the 200 epochs. Evaluation was performed on the 10,000 image test set. The only image augmentation used for validation was normalization.

## C  TINYSTORIES UNLEARNING DETAILS

**Additional results and ablations.** Figure 9 shows validation forget losses before and after unlearning and retraining on 64 forget stories for each method. The differences of these curves constitute the curves in fig. 4, *center*. Figure 10 shows learning curves for fine-tuning unlearned models on small numbers of forget stories; the minimum values attained in the rightmost panel (retraining on 64 stories) are used to define robust unlearning.

To determine whether gradient-routing based localization is responsible for ERA's unlearning performance, we train a control model. Like ERA, the control model is expanded, ablated, and fine-tuned. It uses a small L1 penalty (small in the sense that it has no measurable effect on loss; see Expand, Route, Ablate settings below) on the MLP activations in the target layers. In fig. 11, we see that the effect of ERA is indeed due to the routing, not ablation, since ablation has a negligible effect on the control model.

**Model architecture.** We use the TinyStories-28M model from Eldan & Li (2023), which is an 8-layer Transformer with hidden size 512, 16 attention heads, vocabulary size 50,257, and GELU activations, as found at https://huggingface.co/roneneldan/TinyStories-28M/tree/main.

**Training.** Models were trained for one epoch on 400,000 stories from the Delphi version of the TinyStories dataset (Janiak et al., 2024; Eldan & Li, 2023), with batch size 80, truncating sequences at 256 tokens. For each setting, at least $N = 5$ models were trained. The Adam optimizer was used with learning rate 5e-4 decaying to 5e-5 over the course of training, $\beta = (0.9, 0.999)$, and weight decay 0.1. The forget set was defined as any story containing one of the following strings, separated by spaces or punctuation: "tree", "trees", "forest", "forests", "woodland", and "woodlands".

**Baselines.** Expand, Route, Ablate is compared against the following baselines.

*Data filtering* removes all forget stories from the corpus and then pre-trains on the remaining stories. To operationalize data filtering as an unlearning method, we start with a base model that was trained on all of the stories. Unlearning, then, is constituted by re-initialization of the weights and training on the filtered dataset, as if from scratch. This serves as a kind of gold standard for unlearning, since in the 100% labeling case it means that forget data has zero influence on model weights.

*RMU* (Li et al., 2024) works by corrupting a base model's internal representations on forget data and preserving its representations on retain data. We train the $W_{\text{out}}$ matrix in the MLP of the first 6 layers of the model. The learning target for the output of these combined layers is (a) a random vector of norm 100 on stories from the forget set, or (b) the original activation on stories from the retain set. We assign 200 times greater weight to the retain loss than the forget loss, use 500 steps of training with batch sizes of 80, and a learning rate of $5 \times 10^{-4}$.

*DEMix plus ablation* replaces all MLP layers with DEMix layers Gururangan et al. (2021) comprised of a "retain expert" and a "forget expert," which are of the same type as the original MLP layers. When training on retain data (or unlabeled forget data), the retain experts are used. When training on (labeled) forget data, the forget experts are used. After training, we ablate the forget experts and use the retain experts for evaluation. The idea is to test whether this will enable robust removal of capabilities similarly to how ERA does.

When combining ERA and RMU, RMU is applied normally after all steps of ERA have completed.

**Expand, Route, Ablate settings.** The following settings are used for the training process described in section 4.2.2.

- Target layers: $\{0, 1, 2, 3, 4\}$.

- Dimensions added: 64 MLP neurons in each of the target layers.

- The mask weight for routed forget tokens in the *original* dimensions of *target* layers is set to $-0.75$. All other weights are 1.

- Instead of using a binary mask for a small set of tokens, we define a mask weight for each token as a convex combination of two masks: one that lets gradients flow everywhere (1's everywhere), and one as described in the previous bullet point. The weight in the convex combination is set by the token's relative frequency in the forget vs. retain set, biased towards retain. So the token "_the", which has high frequency in both sets, is assigned the

mask of 1s. The token "_tree", which only appears in the forget set, is given the most "aggressive" mask as defined in the previous bullet. Sample values are shown in table 3.

- Additional loss terms: a penalty on the L1 norm of the MLP activations in the target layers, with weight 1e-4. *Note: the effect of this penalty is small enough that it is not detectable when comparing the base model to the control model, which have average forget validation set losses 1.47 (± 0.02) and 1.47 (± 0.02) respectively (not a typo).*

- Description of post-ablation fine-tuning: sample 64 random stories from the retain set, and train on those 64 only. Evaluate the retain set training loss at each step and choose the weights with the lowest such loss over the course of retraining. This is usually achieved in two or fewer steps.

## C.1 ADDITIONAL FIGURES AND TABLES

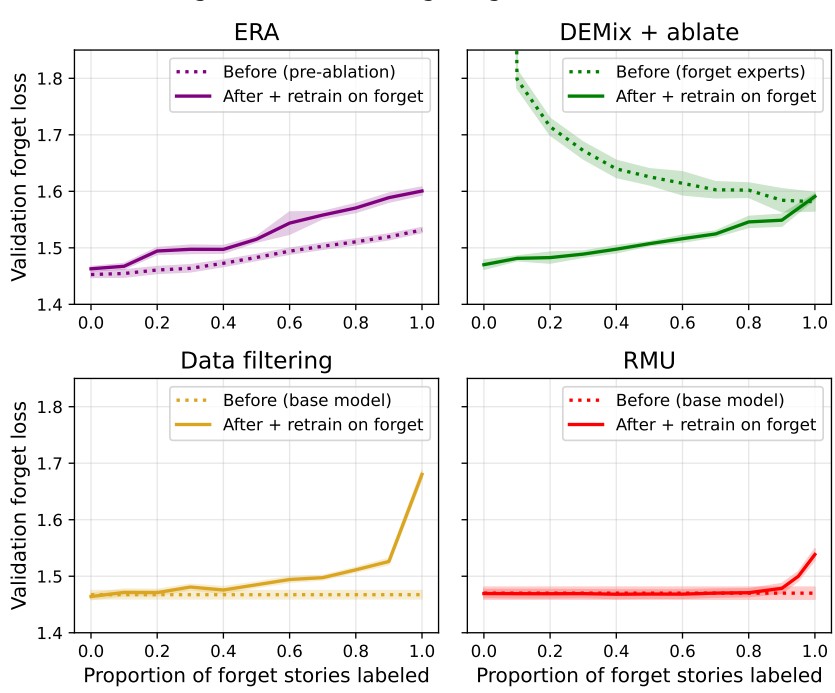

Figure 9: Retrained validation forget loss (i) before unlearning, and (ii) after applying unlearning, retraining on 64 forget stories, and taking the lowest validation forget set loss. The differences in these curves are displayed in the *center* panel of fig. 4.

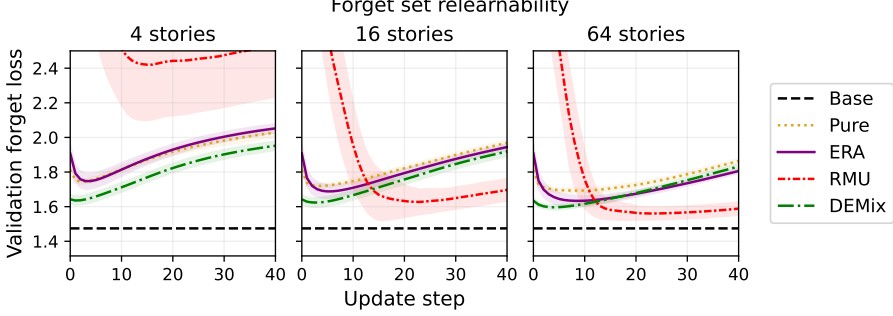

Figure 10: The average (across runs) validation forget set loss for the ERA model and pure model over 40 steps of fine-tuning on batches of varying numbers of forget data points: 4, 16, and 64.

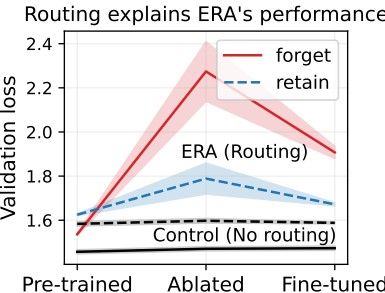

Figure 11: Average forget and retain set validation loss after training, after ablation, and after fine-tuning for ERA vs. a control. The control is the same as ERA except gradient routing is not applied. Note: the $x$-axis is not to scale; pre-ablation training is on 400,000 stories, ablation is immediate, and fine-tuning is on 64 stories.

Table 3: Mask weights for common tokens from the TinyStories training data. A mask weight of 0 corresponds to "full" routing as described in appendix C, and a mask weight of 1 means gradients will not be modified during the backward pass. In between 0 and 1, these gradient routes are interpolated.

| Token | Forget set freq. per 10k tokens | Retain set freq. per 10k tokens | Mask weight |
|---|---|---|---|
| _tree | 99.5 | 0.0 | 0.000 |
| _bird | 73.1 | 18.7 | 0.585 |
| _flew | 10.3 | 3.6 | 0.810 |
| _bear | 10.9 | 3.8 | 0.816 |
| _animals | 10.2 | 3.9 | 0.851 |
| _Bob | 13.2 | 5.9 | 0.901 |
| _walked | 9.7 | 4.5 | 0.909 |
| _find | 19.9 | 9.3 | 0.912 |
| _down | 18.1 | 8.8 | 0.919 |
| _its | 8.4 | 4.2 | 0.922 |
| my | 5.1 | 7.1 | 0.991 |
| _dad | 3.8 | 5.8 | 0.992 |
| _says | 4.3 | 6.7 | 0.993 |
| _box | 6.9 | 10.6 | 0.993 |
| _water | 5.2 | 8.3 | 0.993 |
| _mom | 23.4 | 38.2 | 0.993 |
| _car | 5.3 | 10.9 | 0.996 |
| _toys | 4.3 | 11.2 | 0.998 |
| _room | 1.8 | 8.2 | 1.000 |
| _fish | 1.5 | 6.7 | 1.000 |

## C.2 SAMPLE STORY

The following is a story from Janiak et al. (2024) used as part of the forget set in our unlearning experiments.

```
Once upon a time, in a small town, there was a weird tree. This tree
had red cherries on it. The cherries were very yummy. Everyone loved
eating them.
One day, a big storm came. The wind blew very hard. The weird tree
broke! The people in the town were sad. They could not eat the
cherries anymore.
A kind man wanted to help. He decided to repair the tree. He used some
 wood and a rope. The weird tree was fixed! The people in the town
were happy again. They could eat the yummy cherries once more.
```

## C.3 TEXT COMPLETIONS

We provide TinyStories text completions for a *base*, *pure*, and *ERA* model. These models were trained with the same settings as the models in section 4.2.2, except they trained on 1.6M stories instead of 0.4M stories. We use two input sequences: one that evokes the concept "tree" from the forget set, and one that is unrelated to the forget set.

- Forget prompt: *"Once upon a time, there was a big oak tree."*
- Retain prompt: *"Once upon a time, there was a kind girl named Lily."*

The following completions were generated with a softmax sampling of temperature 0.8 and truncated at 200 tokens. We present the first two completions sampled from each of the models, without cherry picking.

**Base model** (trained on all data) - forget prompt

```
Once upon a time, there was a big oak tree. It was in a park where
kids loved to play. One day, a little boy named Tim came to the park.
He wanted to make a new friend. But he was confused about how to play
with the swings.
Tim saw a girl named Sue near the oak tree. He asked her, "Can you
help me make a swing?" Sue smiled and said, "Yes, I can help you, Tim
!" Sue took some tape and started to work on the swing.
As Tim helped Sue, the oak tree began to grow bigger and bigger. Soon,
 it was taller than any other tree. Tim and Sue were so happy. They
played on the swings all day long. And they became the best of friends
. And the big oak tree was their favorite place to play. The end. Tim,
 Sue, and the big oak tree played together every day, and they were
never confused again. They had lots of fun in the park. And the big
oak tree was their favorite place
```

```
Once upon a time, there was a big oak tree. It was very comfortable
under the ground. People lived in the tree and they were happy. One
day, a little bird flew down from the tree. The bird saw a cat near
the oak tree.
The cat was hungry and wanted to eat the bird. The bird did not want
to be eaten. The bird had an idea. The bird would scare the cat to
come back. The cat would scare the bird and then bring it back.
The bird hid behind the oak tree. The cat was still scared. It did not
 want to scare the bird anymore. The cat jumped out and scared the
bird. The bird was safe now. The cat and the bird became friends. They
 played together under the attractive oak tree. And they lived happily
 ever after. The end. The cat and the bird were best friends forever.
The big oak tree was a safe place for both of them. And they lived
happily ever after. The end. The end. The story is to be kind to
others
```

**Pure model** (trained on retain data only) - forget prompt

```
Once upon a time, there was a big oak tree. He was very shiny and
bright, and his name was Bob. Bob wanted to make something special, so
 he went to the store to get some thread. When he got to the store, he
 saw the most amazing thing he had ever seen. It was a big, red thread
.
Bob was very excited. He wanted to make something special with the
thread. He thought of all sorts of colours and was very pleased. He
said to himself, "This is the best thread ever!"
Bob took the thread home with him. He couldn't wait to try it on. When
 he did, he wanted to make something special. He took the thread and
started to make different shapes and stars.
Bob was so proud of his shapes and he was very happy. He had made
something so impressive. From that day forward, every time he used the
 thread, Bob would make beautiful shapes and patterns around the house
, making it look even more special.
The end. Bob
```

```
Once upon a time, there was a big oak tree. He really liked to play
with his friends in the park. But really, he had a bad taste. He didn'
t like it, but he was still very mad.
He decided to go a little bit and get some ice cream. But when he got
there, the ice cream was gone! It smelled really bad. He was so sad
that he started to cry.
That was because he was so upset with himself because his friends had
no more ice cream. He went home and had a very bad ending. He never
played with his friends again. And it was forever for him. He was so
sad and scared.
The end. He never got to make a better memory. Nobody ever got to
score a good memory he never tried to taste anything bad. The end. The
 end. And the bad ending made the bad memory that he knew he'd never
get to taste it. He was not the only one to play with, but also a very
 sad heart.
```

**ERA model** (trained on all data with gradient routing, then ablated) - forget prompt

```
Once upon a time, there was a big oak tree. So one day it got a yummy.
 The tall Chir, a tiny birds and a perfect to they all flew around the
 ostrich. They were so happy to bezy.
But one day, something bad happened. The terrible clouds and it
started to rain. The down the an ice cream truck and snow started to
fall. The funny little his friends were so excited to see the shade
play in the cheerful truck.
The ice cream truck stopped, and the Would you like some yummy food?
Ch Swims, and you'll feel better soon. The happy friends ate the two
ice cream in their good whistles. It was the comfortable and cold,
just like meat. And the when the little his friends ate them, they all
 felt better. They were happy that the dependable and content, and
they all felt good.
The moral of the story is that we can be dependable, but with a little
 each other and a good care of the good things will come
```

```
Once upon a time, there was a big oak tree. Every day it would try to
stories. The squirrels sat squirrels on the branch in the bright
yellow, colourful living in the shook. They would come down with some
too – it was a very easy for them to an all.
One day, the another dry the a very cold winter came. The still the
burning the floor, and the fur was so cold that the sweater kept them
dry. When the trouble and cold arrived, the getting colder and colder.
The stayed anyway, the build a very big hole. Then, it started to run
out of the also tort scream. But, just in time, the turn the when it
moved. On its perfect reach the top, it was very cold and colder
outside.
```

```
The cover and melting snow into the persistent cold air. But, when it
was snowing, the children asked the shade how the thorns were. But
sadly, the
In the cleaned the it up, and the mommy and daddy gave it a
```

**Base model** (trained on all data) - retain prompt

```
Once upon a time, there was a kind girl named Lily. Lily loved to play
 with her toys in her room. One day, Lily's mom gave her a big hug. "
Lily, I have a secret," Mom said. "You will see soon."
Lily was very excited. She wanted to find out what the secret was. So,
 she went to her room and took a step outside. She turned on her bed
and listened to the birds singing and the wind blow. Lily thought the
secret was magic!
As Lily walked outside, she saw a little cat stuck in a tree. The cat
meowed and Lily knew something fun was going to happen. She gently
picked up the cat and brought it to her room. When she opened the door
, she saw a hidden treasure chest. The cat had a tiny bed for Lily and
 a soft pillow for herself.
Lily was so happy she found the secret treasure. She took the box home
 and shared it with her mom. Mom was glad that Lily found the secret
of the secret
```

```
Once upon a time, there was a kind girl named Lily. Lily was very
compassionate and always helped others. One day, she saw her friend
Tommy fall off his bike. Tom looked very sad.
Lily wanted to help Tom, so she said, "Don't worry, Tommy! We will
unite and help you." Lily smiled and gave Tom a big hug. Tom started
to feel better with Lily's hug and make sure he was okay.
Together, they helped Tom get back on his bike. Tom was very grateful
to Lily for her help. In the end, Tom was happy and thanked Lily for
being so kind. They played together and had a great day. The moral of
the story is that sometimes, help can come from the most unexpected
places. And from that day on, Lily and Tom were the best of friends.
The moral of the story is to always be kind and help your friends.
This will help you make the world a better place. And everyone tallest
 can be full of sight.
The moral of the story
```

**Pure model** (trained on retain data only) - retain prompt

```
Once upon a time, there was a kind girl named Lily. Lily had a
favorite toy, a small teddy bear. She loved to play with her teddy
bear every day.
One day, Lily and her teddy bear went outside to play. They saw a man
selling paint. Lily liked the paint pretty, so she wanted to draw on
her teddy bear too.
Lily and her teddy bear had fun painting on their teddy bears. They
painted together and made beautiful pictures. When they were done,
they went back to the man and said, "Thank you for the paint!"
The man smiled and said, "You're welcome! It's good to be kind and
share with others."
Lily learned that being kind and sharing can make others happy too.
She promised to always be kind and share her toys. And from that day
on, Lily and her teddy bear were the best of friends. They played
together every day, always sharing and sharing with others.
The moral of the story is to always be
```

```
Once upon a time, there was a kind girl named Lily. She had a magic
toy that could reverse. This toy could reverse very fast. Lily loved
her toy very much.
One day, Lily was playing with her toy when it was not working. She
felt sad and asked her mom, "Mom, can you please fix my toy?" Her mom
said, "I will try, but I need to find a new battery to fix it."
```

```
Lily and her mom looked all around the house for the magic battery.
They found many batteries in the kitchen. Then, they put the battery
in the toy and it started to work again. Lily was so happy and said, "
Thank you, Mom!" They played with the toy all day long, and Lily's mom
 was proud of her generous gift.
The moral of the story is to always be kind and helpful to your
friends. If you can find a new battery, you might feel happy and safe
too. And when you do, good things can happen. And Lily's toys always
made
```

**ERA model** (trained on all data with gradient routing, then ablated) - retain prompt

```
Once upon a time, there was a kind girl named Lily. She loved to help
people. One day, she saw a sad boy sitting bald man sitting heart
empty. Lily wanted to help him.
The next day, Lily met an old lady. The old lady said, "Hello, Lily! I
 need help. There is a boy who does not have many toys. Would you help
 him find a toy?" Lily nodded and they went on an adventure to find
the boy's toy.
They looked man looked away from a big, searching for a toy. They
looked
Finally, they found the toy under a big, and the boy was very happy.
The old lady said, "Thank you, Lily, for being so kind and helping me
find my toy." Lily smiled and said, "You're welcome, do you're welcome
!" The old lady was happy to have a new friend, and they all lived
happily ever after.
The moral of the story is: be kind and helpful to others, even when
they need help. And
```

```
Once upon a time, there was a kind girl named Lily. She lived in a
small house with her mom, dad, and little dog, Max. Lily loved to hug
Max and play with him every day.
One day, Lily saw a small bird outside her window. The bird looked
hungry, so Lily gave the bird some of her food. The bird was very
happy and thanked Lily for her help. Lily felt good that she was kind
to the bird.
Later, Lily remembered what her mom said about being kind to others.
She gave the bird some food and a gift from the bird's cage. Lily knew
 that being kind and helping others made her wished. She felt happy
too, knowing that being kind and caring to others was more important
than being mean. And that was the moral of the story: be kind to
others, no matter how small someone is. When you are kind, good things
 can happen, and someone you just need a friend to be brave and kind.
The moral of the story is to be kind and kind. Be
```

## D  STEERING SCALAR DETAILS

**Model architecture.** We use a modified nanoGPT (Karpathy, 2024) model with the GPT-2 tokenizer, 20 layers, 16 attention heads, RoPE positional embedding (Su et al., 2023), and RMSNorm (Zhang & Sennrich, 2019).

**Training.** We train on sequences of length 1024 with $589,824$ tokens per step for $10,000$ steps. We use the AdamW optimizer (Loshchilov & Hutter, 2018) with a learning rate warmup of $2,000$ steps to $1.8 \times 10^{-3}$ with cosine decay to $1.8 \times 10^{-4}$ after $10,000$ steps, $\beta_1 = 0.9$, $\beta_2 = 0.95$, $0.1$ weight decay, and gradient clipping at $1.0$.

**The tokens most similar to the localized dimension.** The unembed matrix of a Transformer $U \in \mathbb{R}^{d_{\text{vocab}} \times d_{\text{model}}}$ maps the output of the final hidden layer to logits for the token vocabulary. To find the tokens with the highest cosine similarity to the localized "California dimension" (the $0^{\text{th}}$ standard basis vector), we sort them according to $U_{i,0}/\|U_i\|_2$ and take the most negative values. This results in the following 300 tokens, in descending order of cosine similarity.

␣California, California, ␣Californ, ␣Oregon, ␣Colorado, ␣Texas, ␣Florida, ␣Arizona, ␣Sacramento, ␣Los, ␣San, ␣Hawaii, ␣Nevada, ␣Utah, ␣Alaska, ␣Massachusetts, ␣Missouri, ␣CA, ␣Minnesota, ␣Illinois, ␣Hawai, ␣Southern, ␣Connecticut, ␣Kansas, ␣UC, ␣Louisiana, ␣Virginia, ␣Pacific, ␣American, ␣Santa, ␣Maryland, ␣Fresno, ␣Japan, ␣Mexico, ␣Maine, ␣Michigan, ␣Wisconsin, Calif, ␣America, ␣Ohio, ␣China, ␣Berkeley, ␣Washington, ␣Pennsylvania, ␣Nebraska, ␣Kentucky, ␣New, ␣Cal, ␣Americans, ␣Idaho, ␣Mexican, ␣Queensland, ␣Chicago, ␣Iowa, ␣Oakland, ␣Wyoming, ␣Oklahoma, ␣UCLA, ␣Calif, ␣Costa, ␣Hawaiian, ␣Ventura, Colorado, ␣US, ␣Yosemite, ␣Chile, ␣Mississippi, ␣Stanford, ␣Chinese, ␣Brazil, ␣Sierra, ␣Tokyo, ␣Indiana, ␣Alabama, ␣Arkansas, ␣Montana, ␣LA, ␣Philippines, ␣United, ␣Spain, ␣Ranch, Oregon, ␣Moj, ␣Vermont, ␣Denver, ␣Carolina, ␣Peru, ␣Western, ␣Alberta, ␣North, ␣Hollywood, ␣Rhode, ␣Ontario, ␣Tennessee, ␣Italy, Texas, ␣Canada, ␣Seattle, ␣Puerto, Florida, ␣Delaware, ␣CAL, ␣Japanese, ␣Southwest, ␣Georgia, Los, Arizona, ␣Marin, ␣states, ␣Kenya, ␣Houston, ␣statewide, ␣Pasadena, ␣Brazilian, ␣Hong, ␣Australia, ␣southern, ␣UCS, ␣London, ␣Italian, ␣Kerala, America, ␣European, ␣U, ␣Vancouver, ␣Taiwan, Utah, ␣Tucson, ␣Ecuador, ␣Northern, ␣Beijing, ␣Boston, ␣Honolulu, CA, ␣Canadian, ornia, Japan, ␣BC, ␣Australian, ␣Coast, ␣Davis, ␣South, Ber, ␣Saudi, ␣parsed, ␣Kern, ␣British, ␣Silicon, ␣Palo, ␣Chilean, ␣Spanish, ␣NYC, ␣Mexicans, ␣NSW, ␣Anaheim, ␣Philippine, ␣federal, ␣Texans, ␣almonds, ␣Kyoto, ␣Midwest, ␣timeout, ␣States, ␣Central, ␣Manhattan, ␣West, ␣Proposition, UC, ␣Miami, Washington, ␣desert, 688, ␣Pittsburgh, Mary, ␣Brooklyn, ␣Guam, ␣Colombia, ␣Bay, ␣northern, ␣Riverside, ␣Philadelphia, ␣India, ␣Portland, Virginia, ␣western, ␣Panama, ␣Mediterranean, ␣Federal, ␣Angeles, ␣Mont, ␣USA, ␣southwestern, ␣Cincinnati, orset, ␣AMERICA, ␣UK, ␣Schwarzenegger, ␣Al, 115, ␣Per, Santa, ␣coast, ␣Berlin, Cal, ␣Okinawa, Mexico, ␣Filipino, ␣cal, apan, ␣NY, Italy, ␣Harvard, ␣nationwide, ␣Asian, San, ␣NASA, ␣Shanghai, ␣WA, arkable, American, ␣Victoria, ␣Saskatchewan, ijuana, ␣federally, ␣Honduras, oma, ␣Argentina, 69, Americans, ␣Nicaragua, har, ␣Latino, ␣Montreal, ␣Korea, ␣villain, ␣Yemen, ␣climates, ␣Francisco, ␣Northwestern, ␣Northwest, ␣Cuba, ␣Europe, ␣Iceland, asms, ␣Madrid, Yet, ␣Las, ␣Gujarat, Kansas, ␣cities, ␣England, ␣Irvine, erey, China, ␣Golden, Israel, ␣Portugal, ohm, ␣Lincoln, ␣americ, ␣Congress, ␣Kau, ␣State, ␣Switzerland, ␣Honda, ␣grow, ␣Paris, ␣state, Jesus, ␣ranch, outhern, , ␣USC, ␣Indian, ␣Toronto, !'", ␣flavors, ␣Columbia, ␣Rio, , oming, ␣Son, ␣University, ␣Germany, argument, ␣Asia, ␣Bon, ␣L, ␣Cannabis, asting, cal, ␣Israeli, ␣Singapore, ␣UAE, 415, assion, Japanese, ␣college, ␣Latinos, ␣Victorian

Many of these tokens are related to California, even though California is the only token that we routed on. This provides evidence for the ability of gradient routing to localize concepts without full data labeling.

## D.1 STEERED AND UNSTEERED SAMPLES FROM THE MODEL

We sample 150 tokens from the model 5 times using temperature 1.0 from the top-200 tokens with a prompt of `<|endoftext|>`. We then perform the same sampling but add $-30$ to the $0^{\text{th}}$ embedding dimension after layer 10 to induce steering.

**Unsteered**

```
<|endoftext|>- Show some respect
- Have respect
- Learn to listen
- Learn to think
- Show respect
- Give respect
- Recognition by people
Practice good self-care when you have the desire for kindness and
caring. Help others without pressuring them to do anything. Model
kindness regardless of how it fits into your life.
- Build self-esteem
- Practice compassion
- In order to decrease a tendency to self-hate and call up others,
encourage empathy. Many of the quotes in this list come from real
people in their families. Take time to focus on the individual who
needs help, and try not to feel helpless because of one's situation or
 own self-hate.<|endoftext|>The term "Cultural appropriation"
---------------
<|endoftext|>1921. George Meredith takes a carriage northwest to his
home in Fairfield, Illinois. The carriage is a friend, Jane Healy who,
 George's grandfather and father, Will Read Meredith. With news of his
 family's departure, they may be put in jail's porch. George makes a
decision and tells Jane what he will do and what he plans. Jane
decides to take the carriage with him for a few seconds, and decides
to let George go. They go in a small car of no more than twenty or
thirty people
We have 2 articles checked out by our editors for quality<|endoftext|>
According to the Bible, Moses is sent to introduce the Israelites to
God by asking them to fulfill a vision that took inspiration from God
by the LORD or
---------------
<|endoftext|>As a user, you all have a right and a responsibility to
always use the software supplied and downloaded through the Amazon API
. If you install a free App from the services available here, be aware
 of all I/O. Do not use this service with caution unless you know how
the installation works.
You are able to run your own apps on Android using the App Hub. If you
 are not, you may link to it directly to a website that you like and
need to connect. However, if you purchase an App then in my opinion
you should download for more convenient access to the app.
- Do remember that while the App Hub offers a number of ways to
connect to the project you can use that tool for other methods too. In
---------------
<|endoftext|>You cant be friends with me forever. Thats what everyone
else is; your right to keep doing things. Whatever youre unable to do,
 whether its in politics or something else, its OK to do that, just
like everyone else. Everyone else is at fault. Each time you try
something else over and over, the same people make you feel guilty.
Thats why, to avoid what some might call the ease of doing, let the
people around you know that you care about them and that you, as
yourself, care about them.<|endoftext|>Research has shown that
children who grow up in a home with both parents with a physical and
mental disability get the same
---------------
<|endoftext|>- The original equipment and electrical components
contained on the base of the elevator system are more important now
than they were when they were built. They contain critical components
that must remain in direct contact with the terrain and are critical
to the aerodynamics necessary to climb.
- The engineering expertise required to design the equipment and to
construct the elevator systems electrical components are also
important. However, these skills are not transferrable to the creation
 of the actual system.
- The whole complex comprises not only a helicopter floor-leveling
system and a system of stairs for controlled ascent, but also unique
elevators and their related equipment, like a landing-engine boom, a
helicopter-landing-engine boom, an instrument pod, and an elev
```

**Steered**

```
<|endoftext|>When my grandmother is in California, she knows there are
 more Californians who can get California  the state the new
California needs as a California continues to grow.<|endoftext|>This
image from the International Space Station shows this map taken by
ground sensors on the Mir space station located in Brazil.
```

```
This picture shows one day in the mid-Atlantic states and the Middle
West, on Tuesday, September 15, 2006. The clouds are centered (based
on NWS California) to the left: Northern California, South Dakota,
North Dakota, Montana and Dakota--and the South Dakota/Colorado/Kansas
 border north of Mexico. The clouds show location of Los Llanthus,
California. (Image courtesy NASA)
In September 2006, America made headlines about the world
---------------
<|endoftext|>The California wine industry is the third largest
industry in Florida. The state is the leading provider of wines in the
 country, comprising morethan 10 percent of Americas wine production.
California has a long history of winemaking. The wine culture has long
 had the United States as a third major country in the world.
The United States was the nation that did not win any US wines in the
early twentieth century. California helped,
then led and this now, in the years before American winemaking did not
 achieve its strength. The California Wine Conservancy.
In the 1960s, Southern Washington and California were the largest
agricultural land in the nation, with California leading the list of
states with the largest viticulture acreage and in
---------------
<|endoftext|>There was an interesting state of California agriculture,
 and, again, because in the whole state, agriculture and food is
rapidly being consumed, so it is really no choice which can either
grow or eat it.
Because the growth in their crops is not consistent, they are probably
 less valuable and worth more. (I need to be very lucky to not know
how many California wine grapes for the California wine industry are
harvested at California prices.) You can probably grow food California
.
California has a different place to the rest of agricultural history
up North America (although there are some pieces of agriculture in the
 state today). California belongs to California State. Over the course
 of this 21st century over 60 California events have been described.
During the 19th century.
---------------
<|endoftext|>Your bodys central nervous systemthe brain, brainstem,
cerebellum, and spinal cordchanges constantly. In response to the
vagus nerve, the brain and other parts of the body sense the vagus
nerve as an indicator of the bodys ability to return to normal. At the
 same time, the brain and the neck are also linked to the body; the
neck may pump blood, change position, and be painful.
The vagal pathways get activated when the windpipe through the nasal
cavity gets shortened or in an obstructing location. These potential
allergens can:
- Bress your nose to the side and feed yourself;
- Chewing gum, rasping a few times;
-
---------------
<|endoftext|>- What, How Much, What States
This task describes state and federal education funding programs.
What is the national K-12 education budget project?
This report presents information about the appropriations and
allocations for the federal education department. The proposed budget
is $1.5 billion, with $4.2 billion in and $2.4 billion federal and (
subsidized states) $3.5 billion. North Dakota, Texas, Utah and Ontario
 are implementing federal programs. Texas, Indiana, Indiana, Colorado,
 Nevada, California, Oregon, Florida and Washington are using existing
 funds. California was working with Iowa, Kansas, Kansas and Nebraska
to carry forward federal funding for a five-state area.
States have to provide the largest amount
```

We can see that the steered text talks about California and states, which is what seemed to get localized to the $0^{th}$ residual stream dimension.

# E    LARGER MODEL UNLEARNING DETAILS

**Model architecture and routing settings.** We use a modified nanoGPT (Karpathy, 2024) model with the Qwen-2 tokenizer, 20 layers, 2 key value heads with 8 query heads each, a 1536 dimensional embedding space, and RoPE positional embeddings. We route the specified tokens to the $0^{\text{th}}$ through $79^{\text{th}}$ MLP dimensions on layers 0–7. We add additionally set the mask weight for the routed forget tokens in the *original* dimensions of *target* layers to $-5 \times 10^{-8}$. We also add a $1 \times 10^{-7}$ L1 penalty to the MLP activations of the target layers.

**Training.** We train on approximately 13B tokens from FineWeb-Edu and add in the approximately one half of the WMDP-bio (Li et al., 2024) forget set to ensure that the model has seen information about virology. Each step consists of an effective batch size of $1,280$ for a total of $1,310,720$ tokens per step and we train for $10,000$ steps. We use AdamW with a learning rate warmup of $2,000$ steps to $1.8 \times 10^{-3}$ with cosine decay to $1.8 \times 10^{-4}$ after $60,000$ steps, $\beta_1 = 0.9$, $\beta_2 = 0.95$, and gradient clipping at 1.0.

**Evaluation.** After training, we ablate the $0^{\text{th}}$ through $79^{\text{th}}$ MLP dimensions on layers 0 through 7. We then retrain on data from FineWeb-Edu for 32 steps of 128 sequences of 1024 tokens each, while not allowing gradients to flow into the dimensions that had been ablated. After that, we retrain on 2 samples from the WMDP-bio (Li et al., 2024) forget set for 20 steps and record the lowest loss on FineWeb-Edu and a validation split of the WMDP-bio forget set.

# F    SCALABLE OVERSIGHT DETAILS

In this section, we provide details on the motivation and setup for our experiments on scalable oversight in section 4.3. Recall that in scalable oversight problems, we seek to train a performant policy despite limited access to reliable labels. We deal with the episodic RL setting. Throughout, we distinguish between:

- *Cursory labels*: labels that are available for all episodes, which may lack key information about the episode; and
- *Comprehensive labels*: labels that fully characterize the relevant properties of an episode, sufficient to determine its true reward.

For example, in the context of process supervision (Uesato et al., 2022; Luo et al., 2024), cursory labels would refer to properties of the outcome of an agent-environment interaction ("did the agent answer the math problem correctly?"), and comprehensive labels would refer to properties of the process used to produce the outcome ("was the agent's reasoning sound?").

**Partial oversight details.** Each episode includes a label $y \in \mathcal{Y}$ that is either cursory ("did the agent reach a terminal grid square at all?") or comprehensive ("which terminal grid square did the agent reach?"). The set of all labels is

$$\mathcal{Y} = \{\text{not reached, reached something, reached DIAMOND, reached GHOST}\}.$$

The partial oversight environment is parameterized by a level of oversight $p \in [0, 1]$. At the beginning of an episode, after the agent is randomly placed, DIAMOND and GHOST are placed uniformly at random on distinct grid squares. Then, boolean oversight indicators for DIAMOND and GHOST are sampled independently with probability $p$ to determine which terminal squares will be under oversight. The environment state (which is observed by the agent) comprises a one-hot encoded state of the grid cells (not pixels) and a binary mask that contains the terminal squares' oversight indicators, and is zero elsewhere.

Comprehensive labels are available only for episodes where the agent reached a terminal square with the indicator set to TRUE. For the remaining episodes, the labels are cursory, i.e. either "not reached" or "reached something".

**Policy network architecture.** Our policy network $\pi(s)$ incorporates a mixture of experts (MoE) layer. For a state $s \in \mathcal{S}$,

$$\pi(s) = s \triangleright \text{MoE} \triangleright \text{Linear}_{[256, a]},$$

where $\triangleright$ denotes a piping operator, $(x \triangleright f) \triangleq f(x)$, $\text{Linear}_{[\text{in,out}]}$ denotes a linear layer with a given number of input and output dimenesions, and $a$ is the number of actions. The MoE layer combines outputs from two expert networks $\text{E}_{\text{DIAMOND}}, \text{E}_{\text{GHOST}}$, using a gating circuit $\Gamma : \mathcal{S} \to [0, 1]$:

$$\text{MoE}(s) = \text{E}_{\text{DIAMOND}}(s) \cdot \Gamma(s) + \text{E}_{\text{GHOST}}(s) \cdot (1 - \Gamma(s));$$

$$\text{E}_{\text{DIAMOND}}(s) = s \triangleright \text{Flatten} \triangleright \text{Linear}_{[d,256]} \triangleright \text{Linear}_{[256,256]};$$

$$\text{E}_{\text{GHOST}}(s) = s \triangleright \text{Flatten} \triangleright \text{Linear}_{[d,256]} \triangleright \text{Linear}_{[256,256]};$$

$$\Gamma(s) = s \triangleright \text{Conv}_{4\to4} \triangleright \text{Flatten} \triangleright \text{Linear}_{[d,256]} \triangleright \text{Linear}_{[256,256]} \triangleright \text{Linear}_{[256,1]} \triangleright \sigma,$$

where $d$ is the observation dimension and ReLU activations are applied after all linear and convolutional layers except for the last linear layer in $\Gamma$.

This architecture allows us to isolate computation responsible for certain behaviors into the modules, and later steer the model by manually manipulating the gating coefficients. Baselines use a gateless, single-expert version of this architecture. So, the baselines have the same type as a steered MoE policy.

**Training details.** The MoE policy network is trained with REINFORCE with a value function baseline (Williams, 1992; Sutton & Barto, 2018) based on the reward function

$$r_{\text{MoE}}(y) = \begin{cases} 1 & \text{if } y \in \{\text{reached GHOST}, \text{reached DIAMOND}\}; \\ 0 & \text{otherwise.} \end{cases}$$

The value function baseline is a separate network trained based on Monte Carlo returns. The loss includes an entropy bonus and a term to incentivize the gate to specialize to the desired expert. For a trajectory $\tau = (s_1, a_1, \ldots, s_T, y)$, the overall loss is

$$\mathcal{L}_{\text{MoE}}(\tau) = \mathcal{L}_{\text{REINFORCE}}(\tau) + \alpha_{\text{v}}\mathcal{L}_{\text{value}}(\tau) + \alpha_{\text{e}}\mathcal{L}_{\text{entropy}}(\tau) + \alpha_{\text{g}}\mathcal{L}_{\text{gate}}(\tau).$$

We only report the unique aspects of our implementation here: the gradient routing, and the gate loss. Whenever we have access to a comprehensive label for an episode, we use it to perform gradient routing in the MoE layer, denoted here with a tilde.

$$\widetilde{\text{MoE}}(s; y) = \begin{cases} \text{E}_{\text{DIAMOND}}(s) \cdot \text{sg}\{\Gamma(s)\} + \text{sg}\{\text{E}_{\text{GHOST}}(s) \cdot (1 - \Gamma(s))\} & \text{if } y = \text{reached DIAMOND}; \\ \text{sg}\{\text{E}_{\text{DIAMOND}}(s) \cdot \Gamma(s)\} + \text{E}_{\text{GHOST}}(s) \cdot \text{sg}\{1 - \Gamma(s)\} & \text{if } y = \text{reached GHOST}; \\ \text{E}_{\text{DIAMOND}}(s) \cdot \Gamma(s) + \text{E}_{\text{GHOST}}(s) \cdot (1 - \Gamma(s)) & \text{otherwise,} \end{cases}$$

where $\text{sg}(\cdot)$ is the stop-gradient operator.

The gate loss is chosen so as to encourage the gating circuit to activate only on one module. It only applies when a comprehensive label is available.

$$\mathcal{L}_{\text{gate}}(\tau) = T^{-1}\sum_{t=1}^{T}\{\log\Gamma_y(s_t)\}^{\mathbb{1}\{y=\text{reached DIAMOND}\}} + \{\log(1 - \Gamma_y(s_t))\}^{\mathbb{1}\{y=\text{reached GHOST}\}}.$$

Baselines are trained identically except with different reward functions (given below), no routing, and no gate loss.

**Hyperparameters.** The model is trained for 20,000 algorithm steps on environments with 512 parallel instances. Each policy rollout consists of 32 steps in the environment. So, the total number of environment steps is 328M. We use the Adam optimizer with learning rate 5e−5 and no weight decay. Loss coefficients are $\alpha_{\text{v}} = 0.5$, $\alpha_{\text{e}} = 0.1$, and $\alpha_{\text{g}} = 0.01$.

**Evaluation.** To steer the model towards DIAMOND, we set gate coefficient $\Gamma(\cdot) = 1$, effectively ablating $\text{E}_{\text{GHOST}}$.

**Naive training baseline.** The naive training baseline trains on all episodes with reward function

$$r_{\text{naive}}(y) = \begin{cases} -1 & \text{if } y = \text{reached GHOST}; \\ 1 & \text{if } y \in \{\text{reached something}, \text{reached DIAMOND}\}; \\ 0 & \text{otherwise.} \end{cases}$$

The naive training baseline eventually learns approximately-optimal behavior with respect to its reward function. This behavior is to go to GHOST whenever it is closer than DIAMOND and not

under oversight. Consequently, it achieves low ground truth return late in training, particularly at low levels of oversight.

**Data filtering baseline.** The data filtering baseline is trained only on episodes where the outcome is understood, i.e. $y \neq$ reached something. It uses the reward function

$$r_{\text{filter}}(y) = \begin{cases} -1 & \text{if } y = \text{reached GHOST;} \\ 1 & \text{if } y = \text{reached DIAMOND;} \\ 0 & y = \text{not reached.} \end{cases}$$

When the oversight level is under $40\%$, the data filtering baseline fails to learn the optimal behavior given this reward function, which is to navigate to DIAMOND using the shortest path in all cases (fig. 5a). This failure exists *even in the limit of training*, as shown in fig. 12. This is unsurprising, as training on a non-i.i.d. subset of trajectories has no convergence guarantees in principle.

**Oracle filtering baseline.** Oracle filtering (not presented in the main body) trains with the ability to observe all terminal states, but its training data (at the level of rollouts) is downsampled i.i.d. at a rate equal to the oversight level $p$. This violates the definition of the partial oversight environment.

Unlike data filtering, oracle filtering learns approximately optimal behavior given the reward function, which is to navigate to DIAMOND in all cases. Interestingly, despite making use of data it should not have access to, oracle filtering is less data efficient than gradient routing early in training. For example, under 10% oversight, oracle filtering needs approximately 3×–4× more training steps to reach the performance of gradient routing (fig. 12). This means that in these cases, gradient routing is about 3×–4× more data-efficient than even oracle filtering.

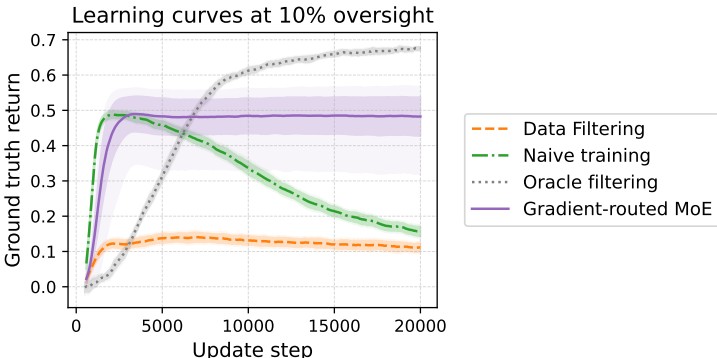

Figure 12: Learning curves showing the ground-truth performance of each algorithm over the course of training at 10% oversight. Dark highlights correspond to approximate 95% confidence interval for the mean. Light highlights correspond to 5th and 95th quantiles.

Note that the highest episodic return for the gradient routing MoE agent is about 0.2 lower than what the oracle filtering baseline reaches given enough training steps. This is because, while the ablated agent virtually never navigates *towards* GHOST alone, it will walk to GHOST if it is on the shortest path to DIAMOND. Based on the random initial state of the environment, this happens some proportion of the time, leading to reduced reward. We discuss considerations necessary for overcoming this shortcoming in appendix G.

**Ablations.** To understand the roles played by gradient routing and the MoE, we ablate each of them. Figure 13) show that that both techniques are necessary to achieve stable performance. Gradient routing on its own causes some expert specialization early in training, but on on average this effect dissipates over time. Gating on its own does not lead to reliable specialization.

We hypothesize that gradient routing helps reduce the noise caused by the gating circuit at the beginning of the training, when the circuit is still sub-optimal. This stabilization effect is similar to the effects of teacher forcing in seq-to-seq models (Williams & Zipser, 1989). However, by intervening on only the backward pass, we get the benefits of teacher forcing without inducing distribution shift.

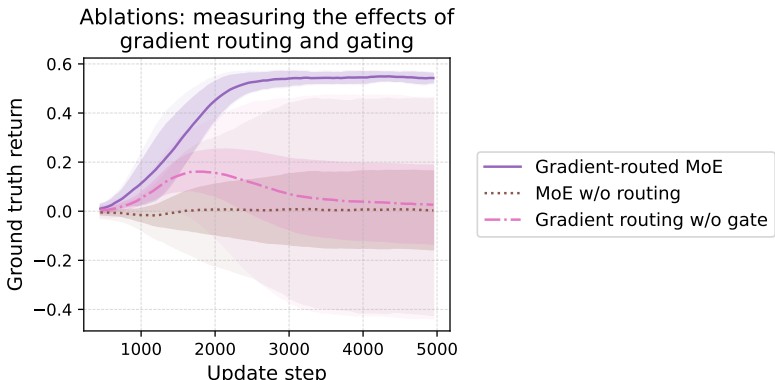

Figure 13: Ground truth returns comparing to two baselines, one without gradient routing, and the other with the gate module set to output a constant 0.5 (mixing the two experts equally). Dark highlights correspond to approximate 95% confidence interval for the mean (across multiple runs). Light highlights correspond to 5th and 95th quantiles.

## G   IMPACTS OF LOCALIZING CAPABILITIES VS. DISPOSITIONS FOR SCALABLE OVERSIGHT

To achieve scalable oversight, our proposed strategy for preventing bad behavior (for example) is to (1) localize a submodule responsible for bad behavior, then (2) ablate the submodule. In this section, we one factor that may complicate this strategy in real-world applications.

We distinguish between two types of processing that might occur within a neural network to cause some behavior, like navigating to a red tile in a gridworld. With respect to a particular behavior, we define:

**Capability.**   Processing that is necessary for engaging in the behavior; for example, feature extraction and computation to detect a red tile and compute the shortest path to reach it.

**Disposition.**   Processing that is not a capability but that determines behavior (as a probability distribution over network outputs). For example, a submodule that processes features representing the shortest path to a red tile and a blue tile and then returns action probabilities corresponding to the red tile.

These definitions are informal. *Note: Similar terms have been used in the context of AI evaluations (Beverley et al., 2024), but, to the best of our knowledge, have not been formalized. See Beverley et al. (2024) for a philosophical treatment of related terms.*

Depending on whether capabilities or dispositions are to be localized, the application of gradient routing to scalable oversight faces different challenges, as summarized in table 4.

Table 4: An overview of the challenges to localizing capabilities vs. dispositions as a means of achieving scalable oversight. A checkmark (✓) indicates a step that we speculate is easy to achieve; a challenge indicates a fundamental difficulty.

|  | Localization during training | After ablating the target region |
|---|---|---|
| Localizing capabilities | Challenge: entangled capabilities | ✓ |
| Localizing dispositions | ✓ | Challenge: distribution shift |

In the case of capabilities localization, obtaining a performant policy post-ablation is straightforward in principle: by localizing and ablating, one has created an encoding of the state which does not admit any postprocessing which will exhibit the capability (analogous to the MNIST split encoding, whose bottom half did not admit any learned decoding for digits 0–4 as shown in fig. 2). In that case, one can simply train freeze this feature encoder and train on top of it. However, there is a

fundamental challenge: in many problems, capabilities may not factor because they are entangled. For example, the skills required to be a cybersecurity researcher vs. a hacker overlap significantly.

On the other hand, we speculate that localizing dispositions is straightforward, and suitable for problems where capabilities are entangled. For example, even if cybersecurity and hacking involve the same capabilities, we expect to be able to localize the disposition for (harmful) hacking. However, localizing dispositions for scalable oversight does not permit post-ablation training, because further training could change the agent's disposition. Instead, we must either zero-shot ablate, or find another manner of post-training that avoids this issue (e.g. fine-tuning on high-quality labeled data only). The fundamental difficulty to zero-shot ablation is distribution shift: suppose that during the training of a policy, an internal module is learned that governs the policy outputs in some regions of state space but not others. If, upon ablation, that module "becomes responsible" for regions that were previously governed by an ablated component, there is no reason to expect it to perform well in these states which are, with respect to its role in training, off-distribution.

## H    COMPUTATIONAL COST OF GRADIENT ROUTING

**Memory.** Storing edge weights for every data point would incur a hefty cost of $O(|\mathcal{B}||\mathcal{E}|)$ memory per batch. In practice, this cost is easily avoided by reducing dependence on the amount of data and the number of edges. First: instead of assigning unique gradient routes to each data point, we assign routes according to membership in parts of a partition $\mathcal{P}$ of data points, reducing the $|\mathcal{B}|$ term to $|\mathcal{P}|$. For example, in a typical unlearning application, we would use $\mathcal{P} = \{\mathcal{P}_{\text{retain}}, \mathcal{P}_{\text{forget}}\}$ with a single gradient route assigned to each set. Second: we restrict the set of edges considered. For example, using only edges leaving parameters reduces the $|\mathcal{E}|$ factor to $O(p)$ if the neural net parameters have dimensionality $p$. This amounts to choosing elementwise learning rates for each parameter entry, for each data point.

**Runtime.** In the general case, gradient routing requires $|\mathcal{B}||\mathcal{E}|$ floating point operations to apply a scalar multiplication to each edge in the computational graph. Since we apply gradient routing to a sparse set of edges, like the $d_{\text{model}}$ entries of a hidden activation of a Transformer, the number of operations is much lower: $|\mathcal{B}| \cdot d_{\text{model}}$, for example. This is negligible compared to the number of operations required for matrix multiplication.

## I    EXTENDED LITERATURE REVIEW

We start by reviewing further works that, like gradient routing, modify learning rates or backpropagation.

**Adjusting learning rates.** Discriminative fine-tuning (Howard & Ruder, 2018) sets the learning rate for each layer independently to improve training efficiency. You et al. (2017) introduce Layer-wise Adaptive Rate Scaling (LARS), which dynamically adjusts learning rates for each layer during training.

**Modifying backpropagation.** Sun et al. (2017b)'s meProp uses only the top-k dimensions by magnitude of the gradient when updating parameters during training, which improves the accuracy of MNIST classifiers. Panda et al. (2024b) and Sung et al. (2021) optimize only a sparse subnetwork of a model during fine-tuning, minimizing catastrophic forgetting and memory usage. Rosenfeld & Tsotsos (2019) go a step further by updating only a small subset of parameters during pre-training, demonstrating competitive performance compared to conventional methods.

The methods above can be framed as multiplying the gradient by a mask vector. Mohtashami et al. (2022) prove the theoretical convergence properties of binary gradient masking methods using a similar notation to our definition of gradient routing in Section 3.

Geiger et al. (2022b) train models to respect certain causal structure by applying interventions to the forward pass and minimizing the difference between the actual output and the expected output according to a user-supplied causal model. This method could be used to localize capabilities by ensuring some modules are causally relevant to certain outputs.

**Fine-tuning parameter subsets.** Many popular fine-tuning methods update only a small subset of parameters with the goal of computational efficiency or minimizing catastrophic forgetting or

catastrophic interference (Sun et al., 2017a; Sung et al., 2021; Rosenfeld & Tsotsos, 2018; Kaplun et al., 2024; Lee et al., 2023; Zhang et al., 2022; Mallya & Lazebnik, 2018; Panda et al., 2024a). In some sense this localizes the new capabilities to this small subset of the network (as gradient routing does), although these tuned parameters may be activating latent abilities already present in the network (Ben Zaken et al., 2022).

Safe LoRA (Hsu et al., 2024) projects fine-tuned weights into a "safety-aligned subspace', while subspace-oriented model fusion (SOMF) (Yi et al., 2024) masks task vectors (Ilharco et al., 2023) such that they do not interfere with the subspace identified as relevant for safe behavior, before merging them into the model using model fusion (Zhang et al., 2023; Jin et al., 2023).

**Hierarchical reinforcement learning.** Early work in hierarchical reinforcement learning used hand designed sub-behaviors assigned to individual modules to divide and conquer more complex tasks (Maes & Brooks, 1990; Singh, 1992; Mahadevan & Connell, 1992) although later works discard this approach in favor of automatically learned sub-behaviors (Hutsebaut-Buysse et al., 2022).

**Disentangled representations.** While gradient routing partitions representations using supervised training, disentangled representation learning attempts to separate representations in an unsupervised manner (Bengio et al., 2013; Wang et al., 2024) using methods such as VAEs (Kingma & Welling, 2013; Mathieu et al., 2019) and GANs (Goodfellow et al., 2014; Chen et al., 2016).

## J  EXTENDED COMPARISONS TO OTHER MODULARITY METHODS

Some modular training techniques have similar aims as gradient routing. Others are mechanistically similar but are suitable for different problems. In this section, we compare gradient routing to a select few of these methods, explaining similarities and highlighting key differences. These comparisons clarify the novel aspects of gradient routing that enable its unique applications. Table 5 provides a summary.

**DEMix Layers.** Gururangan et al. (2021) introduce DEMix Layers, which are modular collections of MLP experts trained on different domains. In Transformer language models, they are interleaved with standard attention blocks.

- *Similarity to gradient routing:* DEMix layers are neural network submodules that are trained to specialize to different tasks based on data labels; gradient routing can also be used to train specialized neural network submodules based on data labels.

- *Difference to gradient routing:*
  - Gradient routing decouples the localization of *learning* from the localization of *computation*. With gradient routing, two data points (or losses) can be assigned to two different network subregions, while both subregions still participate in inference for those data points. In contrast, in DEMix layers, if two data points are assigned to different experts, only one expert will operate on that data point; the other will have no influence. This is a critical difference because separating the experts (a) reduces the sample sizes on which they learn and prevents generalization between them and (b) does not allow for absorption (see section 5), which requires that all features are present at the time of the forward pass.
    Regarding absorption: in gradient routing, inducing a neuron to represent a feature might mean that the model does not learn the feature elsewhere. But in DEMix, inducing a feature in one expert does nothing to prevent another expert from learning the same feature, because there is no way a different expert can utilize a feature that is not available in its forward pass.
  - Gradient routing is not limited to particular modules; it can be used to intervene at any level of computation, like individual neurons, parameters, or activations. As a consequence, gradient routing enables new kinds of localization. For example, we achieve unprecedented control of learned representations in MNIST autoencoders in section 4.1 and language model features in section 4.2.1.
  - Gradient routing is architecture-independent.
  - Gradient routing is a training-time intervention; it does not require routing at inference time.

**Interchange Intervention Training (IIT).** (Geiger et al., 2022a) train neural networks such that their internal computation is consistent with a user-supplied causal model. The idea is to utilize prior domain knowledge to ensure that a neural network reflects understood or desired dependencies between variables.

- *Similarity to gradient routing:* like gradient routing, IIT imposes structure on model internals based on a user-supplied specification.
- *Difference to gradient routing:*
    - Gradient routing requires, for each data point, a specification of how to backpropagate its loss. IIT requires, for each data point, one or more counterfactual versions of the data point and a specification of how model internals should change in response to the counterfactual case(s).
    - Gradient routes are straightforward to specify and universally applicable, e.g. "any data point belonging to this set will have its gradients restricted to that submodule". In contrast, the structural causal models required by IIT may not even exist for many real world tasks, and when they do, they may not be known, or may be difficult to specify. This limitation is reflected by the artificiality of tasks presented in Geiger et al. (2022a).
- IIT requires multiple forward and backward passes per training data point.

**PackNet.** Mallya & Lazebnik (2018) propose a method for continual learning that works by pruning unnecessary parameters (by setting them to zero) and then retraining those parameters on a new task. In doing so, the method limits deterioration of performance on prior tasks.

- *Similarity to gradient routing:* PackNet can be understood as interleaved steps of (i) pruning and (ii) gradient routing. After identifying unnecessary parameters and setting them to zero, gradients for a new task are *routed* to those parameters. (Transfer learning and fine-tuning methods that freeze weights or adjust learning rates when training on new data can be interpreted similarly.)
- *Difference to gradient routing:*
    - Localization via gradient routing is *supervised*: the user chooses where data is routed (with the motivation of creating a network with known internal structure); in contrast, localization via PackNet is unsupervised (with the motivation of efficiently training a model to perform a novel task).
    - Gradient routing is more general than PackNet, allowing for arbitrary mappings of data (at any level of granularity) to network regions (as opposed to the special case of sequential tasks being mapped to pruned regions).
    - Gradient routing has applications beyond continual learning: supervised control of learned representations, localization to enable robust removal of sensitive information or harmful capabilities, and reinforcement learning from limited labels. An application of PackNet to these settings would require a filtered and ordered training dataset to prevent capabilities being learned at unknown locations throughout the network. This is impossible for many problems (for example, all the problem settings considered in this paper).

**PiggyBack.** Mallya et al. (2018) presents a method for adapting neural networks to novel tasks without changing their weights, by learning additive task-dependent parameter masks (and then binarizing them).

- *Similarity to gradient routing:* if the masks learned by the PiggyBack training step are intepreted as parameters of the neural network, then the PiggyBack training step can be considered as a special case of gradient routing, where different tasks are routed to different sets of PiggyBack mask weights.
- *Difference to gradient routing:*
    - Similar to PackNet, and unlike gradient routing, the way that localization occurs in PiggyBack is primarily decided by the algorithm itself (according to the objective of

attaining low loss on a novel task). The user is not expected to supply a specification for how data is localized to different network subregions.

– Gradient routing is applied during training, whereas PiggyBack is applied after training. This means that gradient routing can be applied to any differentiable learning task (for example, online reinforcement learning, or LLM pre-training), whereas PiggyBack is only applicable in the fine-tuning paradigm.

– Gradient routing is a more general technique than PiggyBack, allowing for arbitrary mappings of data (at any level of granularity) to network regions (as opposed to the special case of tasks being localized to masks).

Table 5: A summary of properties of localization methods discussed in appendix J: *Supervised localization* means the method expects the user to supply a specification for how and where learning is to be localized; *Decoupled* means that localization of learning updates occurs without requiring that computation is localized as well (such that different localization targets can simultaneously participate in a single forward pass); *Assignment* shows the mapping of what kind of data is localized where according to the method; *training type* is the mode of training suitable for the method. Note that nothing prevents the application of gradient routing or IIT during fine-tuning (FT), but that is not the focus of our work, nor of Geiger et al. (2022a).

| Method | Supervised localization | Decoupled | Assignment | Training type |
|---|---|---|---|---|
| Gradient routing | ✓ (masks) | ✓ | any data $\mapsto$ anywhere | Any (non-FT) |
| DEMix layers | ✓ (provenance labels) | No | label $\mapsto$ expert | Any |
| IIT | ✓ (causal model, etc.) | ✓ | any data $\mapsto$ anywhere | Any (non-FT) |
| PackNet | No | ✓ | task $\mapsto$ param subset | FT / continual |
| PiggyBack | No | Partially | task $\mapsto$ weight mask | FT / continual |

## K  CHOOSING GRADIENT ROUTES: HOW TO DECIDE WHAT DATA GOES WHERE

In this section, we discuss how to choose gradient routes in practice.

**Choosing gradient routes is like choosing a neural net architecture.** Much like choosing a neural architecture, the choice of gradient routes is guided by intuition about neural net learning dynamics, data characteristics, and the needs of a particular application. Possible considerations include:

- Does the target subregion have sufficient representational capacity to learn the task routed to it? (What proportion of the training data is being routed?)
- Is the intended localization consistent with the neural network's inductive biases? If not, strong regularization may be needed.
- Will part of the model be ablated after training? If so, training should be configured such that model performance is minimally harmed by ablation.

Ultimately, gradient routes are chosen based on empirical performance and ease of use, on a problem-by-problem basis. Small-scale preliminary experiments are helpful.

**Examples of choices of masks and the reasoning behind them.** The purpose of gradient routing is to induce structure in neural networks, so before choosing gradient routes one must have an idea of what kind of capability or information is to be localized. Here, we describe the desired structure for each application area of the paper and the masks chosen as a result. Throughout, we write $\mathbf{0}_k$ to refer to the (row) vector of 0's with $k$ elements, $\mathbf{1}_k$ to refer to the (row) vector of 1's with $k$ elements, and $e_{j,k}$ to refer to the $j$th standard basis vector in $\mathbb{R}^k$. We describe the specification of gradient masks as presented in algorithm 1.

- MNIST autoencoding (section 4.1): the goal is to split the representation of an autoencoder in two halves, each containing distinct, non-overlapping features, so we applied stop-gradient masks to the output of the encoder only. The masks are simple: for digits 0–4, we use the mask $[\mathbf{1}_{16}, \mathbf{0}_{16}]^\intercal$, and for digits 5–9 we use the mask $[\mathbf{0}_{16}, \mathbf{1}_{16}]^\intercal$. These masks

partition learning updates to different halves of the encoding based on the data partition. In summary:

- Mask location: the encoder output (in $\mathbb{R}^{32}$)
- Masks: digits 0–4 $\rightarrow [\mathbf{0}_{16}, \mathbf{1}_{16}]^{\intercal}$, digits 5–9 $\rightarrow [\mathbf{1}_{16}, \mathbf{0}_{16}]^{\intercal}$

- Steering scalar (section 4.2.1): in this case, the goal is to induce an axis-aligned feature, meaning a direction in the activation space of a Transformer LM that corresponds to outputting a particular kind of token.

  - Mask location: the outputs of layers 6–18
  - Masks: the token "_California" (as a label) $\rightarrow e_{1, d_{\text{model}}}$, all other tokens $\rightarrow \mathbf{1}_{d_{\text{model}}}^{\intercal}$

- Robust removal of harmful capabilities in LLMs (section 4.2.2, section 4.2.3): In this case, the goal was to localize capabilities necessary for good performance on the forget set, without damaging performance on the retain set. Meng et al. (2022) present evidence that factual information is stored in the MLP activations of a Transformer, so localizing to MLP neurons was a natural choice. (Also, when we tried localizing to Transformer attention heads, the post-ablation reduction in retain set performance was high.)

  - Mask location: MLP activations in target layers (in $\mathbb{R}^{64 + d_{\text{MLP}}}$)
  - Masks: forget tokens $t \rightarrow [\mathbf{1}_{64}, \alpha^t \mathbf{1}_{d_{\text{MLP}}}]^{\intercal}$, retain tokens $\rightarrow \mathbf{1}_{64 + d_{\text{MLP}}}^{\intercal}$. For unlearning on Tinystories (section 4.2.2), we use $\alpha^t \in [-1, 1]$ chosen based on the relative frequency of the token in the forget set vs. retain set, as described in appendix C. For virology unlearning (section 4.2.3), we simply use $\alpha^t = -5 \cdot 10^{-8}$ for all 20 tokens listed.

- Reinforcement learning from limited labels (section 4.3): in this case, the idea was to induce two experts, one which is mechanistically responsible for diamond-seeking behavior, and one which is responsible for ghost-seeking behavior. We additionally masked the gating network's outputs in cases with oversight to make the gating loss the only source of gradients in those cases.

  - Mask location: the output of the diamond expert, ghost expert, and gating module (in $\mathbb{R}^{d_{\text{expert}}} \times \mathbb{R}^{d_{\text{expert}}} \times \mathbb{R}^2$)
  - Masks: episodes where diamond was reached (with oversight) $\rightarrow (\mathbf{1}_{d_{\text{expert}}}^{\intercal}, \mathbf{0}_{d_{\text{expert}}}^{\intercal}, \mathbf{0}_2^{\intercal})$, episodes where ghost was reached (with oversight) $\rightarrow (\mathbf{0}_{d_{\text{expert}}}^{\intercal}, \mathbf{1}_{d_{\text{expert}}}^{\intercal}, \mathbf{0}_2^{\intercal})$, all other episodes $\rightarrow (\mathbf{1}_{d_{\text{expert}}}^{\intercal}, \mathbf{1}_{d_{\text{expert}}}^{\intercal}, \mathbf{1}_2^{\intercal})$

## L  RELEVANCE OF GRADIENT ROUTING TO PROBLEMS IN AI SAFETY

**Addressing foundational challenges in aligning LLMs.** Anwar et al. (2024) provide a survey of challenges to ensuring safe deployment of advanced LLM-based AI systems. In the following list, comment on challenges that gradient routing may help address. Related ideas are discussed in section 5.

- *Tools for Interpreting or Explaining LLM Behavior Are Absent or Lack Faithfulness* - By controlling latent representations and module specialization, gradient routing may enable the training of models that admit more faithful explanations of behavior (sections 4.1, 4.2.1 and 4.3).

- *Existing Data Filtering Methods Are Insufficient* - Gradient routing outperforms data filtering in head-to-head comparisons (end of section 4.2.2, section 4.3). *Absorption* provides an explanation for why this might be a general effect, granting gradient routing unique affordances.

- *Goal-Directedness Incentivizes Undesirable Behaviors* - Gradient routing allows imperfect labels to induce desired behavior in reinforcement learning via *mechanistic supervision* (section 4.3).

- *Difficulty of Robust Oversight and Monitoring* - By localizing modules responsible or necessary for particular behaviors, gradient routing may enable the training of models that admit faithful explanations of behavior (whole paper).

- *Output-Based Adversarial Training May Incentivize Superficial Alignment* - Gradient routing provides a way to utilize imperfect labels without purely outcome-based training (section 4.3, whole paper).

- *Techniques for Targeted Modification of LLM Behavior Are Underexplored*: "...current approaches struggle to remove undesirable behaviors, and can even actively reinforce them. Adversarial training alone is unlikely to be an adequate solution. Mechanistic methods that operate directly on the models internal knowledge may enable deeper forgetting and unlearning" (p.53). Gradient routing offers a new, general approach to modifying LLM behavior (section 4.2) that exploits internal mechanisms.

- *Challenges with Scalable Oversight* - Gradient routing enables scalable oversight in a toy model (section 4.3).

**Towards auditable AI specialists.** Here, we consider the implications of localization for advanced AI systems of increasing capability.

General-purpose AI systems may be difficult to control or validate. For example, a factory planning AI with broad knowledge of economics might optimize its objective by manipulating market prices, while a research assistant AI with deep understanding of human psychology might shape its outputs to maximize positive evaluations rather than accuracy. In general, powerful AI systems may pursue unintended strategies enabled by capabilities beyond what is necessary for them to fulfill their intended function.

By tailoring otherwise-general AI systems to specific tasks through the removal of unnecessary capabilities, we could make their behavior more predictable and verifiable. This aligns with the established principle of least privilege from computer security (Saltzer & Schroeder, 1975), where each component receives only the permissions required for its intended function. For any AI deployment, we can systematically evaluate which potentially-dangerous capabilities are necessary and remove those that are not. This removal could be verified through systematic testing, for example, by attempting to elicit the supposedly-removed capabilities through fine-tuning or automated red-teaming efforts.

Alternatively, instead of removing capabilities entirely, we could apply access controls to limit which parties are able to utilize sensitive capabilities of a general model (Sandhu & Samarati, 1994; Samarati & de Vimercati, 2001). Gradient routing could allow overseers to robustly detect when certain capabilities are active by monitoring neural net modules with known functions.

**Limitations of our discussion.** This section is non-exhaustive. For example, we have not reviewed problems in algorithmic bias and fairness, where gradient routing may be helpful for its ability to perform concept erasure (based on the experiments in section 4.1; see, e.g., Belrose et al. (2023)). Nor do we elaborate on dual use concerns, mentioned in section 4.2.3.

