# OpenReview forum: "Gradient Routing: Masking Gradients to Localize Computation in Neural Networks"
_ICLR.cc/2025/Conference — Submitted to ICLR 2025_

### Official Review · Reviewer_gjLh · 2024-10-31

**Soundness:** 2
**Presentation:** 2
**Contribution:** 2
**Rating:** 5
**Confidence:** 5

**Summary:**

This paper proposes a gradient-routing training method for isolating capabilities to specific subregions of neural networks. We show that gradient routing can be used to (1) learn representations that are partitioned in an interpretable way; (2) achieve robust unlearning by ablating pre-specified subregions of the network; and (3) achieve scalable supervision of reinforcement learners by localizing modules responsible for different behaviors.

**Strengths:**

1. Based on the gradient, a training method is proposed to isolate the ability to a specific sub-region of the neural network.
2. Gradient routing has implications for safely deploying AI systems, especially in high-risk scenarios where black-box methods are not robust enough.

**Weaknesses:**

1. In Section 3, authors did not clarify the gradient routing, thus make this paper hard to understand. For example, how gradient mask is generated.

2. Authors did not clarify how gradient mask is generated. I hope you can provide the formula or pseudocode for the mask generation at line 178 of the pseudocode.

3. The MNIST dataset has too little data to prove the generalizability of the method. Experiments on more widely used datasets such as ImageNet and COCO can be more convincing.

**Questions:**

1. For representations learning, the gradient routing method can be applied to more common image classification tasks?
2. Can you provide a more detailed description in Section 3?
3. Could you add a Certificate using the top half encoding for comparison in Section 4.1?

---

> ### Author Response · Authors · 2024-11-19
> **Response to gjLh**
>
> We thank the reviewer for acknowledging the implications of gradient routing for the safe deployment of powerful AI problems. We also thank the reviewer for identifying some major opportunities for improvement for the paper, which we have now incorporated. We hope the reviewer finds that we have clarified how masks are generated and established the general applicability of gradient routing.
>
> **Response to weaknesses:**
>
> 1./2. We have added a paragraph to the end of section 3 explaining the general strategy for choosing gradient masks throughout the paper and referencing a newly-added appendix K, which gives precise specifications for masks used for all experiments. We also remind the reviewer that each section contains descriptions of how gradients are routed; for example Figure 2.a. and Figure 3. However, these explanations were not posed in terms of masks, so we are grateful that the reviewer gave us the opportunity to make this improvement.
>
> 3\. The reviewer is right that MNIST MLP autoencoding is a limited setting. We have strengthened this part of the paper by adding analogous experiments on a CIFAR100 ResNet classifier to the appendix, section B.1. These results match the experimental design of MNIST and show comparably strong results. We also encourage the reviewer to consider our results on transformer LMs and reinforcement learning, showing the diversity of gradient routing applications.
>
> **Response to questions:**
>
> Questions 1. and 2. are answered above. We again thank the reviewer for helping us make these improvements.
>
> > 3. Could you add a Certificate using the top half encoding for comparison in Section 4.1?
>
> This is a great idea. We have added a Certificate to the top half encoding and changed the histogram of results to show how each certificate is able to reproduce differents sets of images from different halves of the encoding. We also include reconstructions from the top half encoding in figure 6 in appendix B.
>
> **Summary:**
>
> We are grateful for the reviewer's feedback and hope that we have addressed all the reviewer's concerns here and in the top-level comment clarifying the contributions of our work. We ask the reviewer to consider raising their scores to reflect these clarifications and improvements, or to let us know if they have futher concerns about or suggestions for the paper.

---

### Official Review · Reviewer_Dmtv · 2024-11-01

**Soundness:** 3
**Presentation:** 3
**Contribution:** 2
**Rating:** 5
**Confidence:** 4

**Summary:**

This work introduces gradient routing, a training method that enhances neural network safety by isolating specific capabilities within distinct network subregions. By applying data-dependent, weighted masks to gradients during backpropagation, gradient routing allows users to control which parameters are updated for particular data points. This approach achieves three main outcomes: (1) interpretable partitioning of representations, (2) effective unlearning by ablating targeted subregions, and (3) improved oversight of reinforcement learning by confining behaviors to distinct modules. The results indicate that gradient routing can localize capabilities even with limited data, suggesting its potential for complex, real-world applications.

**Strengths:**

1. This paper is well written and easy to follow.
2. The idea is simple and effective
3. The topic is really important for helping understand neural networks and learned representations.

**Weaknesses:**

1. My main concern is about the contributions. Similar ideas has already been proposed by previous papers
(1) Piggyback [1] and PackNet[2] tries to learn different tasks better with different subsets of the weights by leveraging a binary mask.
(2) Parameter efficient tuning method for Large language models(LLMs) such as lora[3], can efficiently learn different downstream tasks using different adaptor modules.

So the contributions of this paper are not enough.

2. Need more large scale experiments. Given the idea is not that new, only small scale experiments may not enough.



[1] Piggyback: Adapting a Single Network to Multiple Tasks by Learning to Mask Weights
[2] PackNet: Adding Multiple Tasks to a Single Network by Iterative Pruning
[3] LoRA: Low-Rank Adaptation of Large Language Models
[4] S-LoRA: Serving Thousands of Concurrent LoRA Adapters

**Questions:**

1. Clarification of Contributions

 2. Choosing the Routing Strategy:
Since gradient routing relies on predefined, data-dependent masks, an essential question is how to select an effective routing strategy. Could the method involve dynamic routing decisions based on the characteristics of each input, allowing gradients to flow through particular layers or weights? Further clarification on decision criteria for different types of data would strengthen the approach's versatility and practicality.

3. Impact of Initialization and Routing Interactions:
Gradient routing may be influenced by the initialization of weights, particularly concerning the Lottery Ticket Hypothesis [1]. If certain weights are initially unproductive and specific inputs route gradients exclusively to these weights, it could hinder learning.


[1]. The Lottery Ticket Hypothesis: Finding Sparse, Trainable Neural Networks

---

> ### Author Response · Authors · 2024-11-19
> **Response to Dmtv (1/2)**
>
> We thank the reviewer for raising important questions about novelty so that we can clarify them for future readers, and for the insightful connection to the Lottery Ticket Hypothesis. We are also delighted to hear that the reviewer thought the paper was "well written" and that the method is "simple and effective" and "really important" for understanding NNs and representation learning. This is great to hear!
>
> We understand the reviewer has concerns about the novelty of the idea. We have addressed these concerns in the top-level comment (above), explaining that gradient routing has unique properties which enable the novel applications given in the paper. We elaborate below.
>
> **Response to weaknesses:**
>
> > My main concern is about the contributions. Similar ideas has already been proposed by previous papers (1) Piggyback [1] and PackNet[2] tries to learn different tasks better with different subsets of the weights by leveraging a binary mask. (2) Parameter efficient tuning method for Large language models(LLMs) such as lora[3], can efficiently learn different downstream tasks using different adaptor modules.
>
> We thank the reviewer for bringing Piggyback and PackNet to our attention. The training steps of these algorithms can be viewed as special cases of gradient routing, which is a neat connection! (Similarly, we note that methods which train a subset of weights, like transfer learning, or low rank-adapters, can be viewed as special cases of gradient routing too.) As mentioned in the top-level comment, we have added detailed comparisons of gradient routing to Piggyback and PackNet in appendix K. (If the point is not clear, we are happy to add more comparisons.)
>
> However, we emphatically disagree with the suggestion that our contributions (see top-level comment) are redundant with ideas from fine-tuning and continual learning. Although the ideas are superficially similar ("apply gradient updates only to specific parameters"), they are quite different. The methods mentioned by the reviewer are concerned with efficient training to achieve low loss in the fine-tuning regime. In contrast, our contribution is about giving the ML practitioner flexible control over model internals; we offer novel ideas for how this can be achieved and what problems it can be applied to. Furthermore, our method is more general, allowing for kinds of localization not achievable by prior methods. For example, we demonstrate a variety of ways of localizing parameter updates for _individual tokens_ in language models (section 4.2).
>
> We anticipate a possible objection: "yes, the aforementioned methods were designed for fine-tuning and continual learning, but you could have studied them as localization methods." Our answer is twofold. (i) In a sense, we _did_ study them, because our formalization of gradient routing generalizes these methods. (ii) To the best of our knowledge, we are the first to propose localization as a solution to the problems in the paper. We see this as a very important aspect of our contribution, so to some extent it is irrelevant _which_ method for localization we chose to apply (as long as the method works). We have identified novel ways to bring localization to bear on open problems in machine learning, and done so via a simple and general framework.
>
> > So the contributions of this paper are not enough.
>
> We have attempted to provide a thorough and compelling response to this statement. We welcome further questions if anything is unclear!
>
> > Need more large scale experiments. Given the idea is not that new, only small scale experiments may not enough.
>
> We believe our idea is quite new, for the reasons explained above. We also hope the reviewer will consider the breadth and design of our experiments (including newly-added experiments on a ResNet classifier), which we believe convey a lot of useful evidence about gradient routing. We also include all of our code with the paper so that others can replicate and build on the results, further strengthening the contribution.

---

> ### Author Response · Authors · 2024-11-19
> **Response to Dmtv (2/2)**
>
> **Response to questions:**
> 1. We have clarified contributions in the top-level comment and in Response to Dmtv (1/2).
> 2. Regarding the choice of routing strategy: this is a great point. We have added a paragraph to the end of section 3, and an extended discussion in appendix K regarding how to choose gradient routes. Regarding whether the method could involve dynamic routing decisions depending on the input, the answer is that it can, and does! The MNIST autoencoder could be seen as one example. We also applying routing with continuous mask weights based on relative token frequencies in our Tinystories experiments (section 4.2; appendix C has details). However, the gradient routes we've used so far have been relatively simple. In the discussion (section 5), we state:
>     > In our experiments with language models, we route gradients on a token-by-token basis, ignoring neighboring tokens. This naive strategy is surprisingly effective. However, it is plausible that contextual information will be critical in some problems, necessitating routing strategies that depend on entire sequences. Finding practical ways of choosing what data to route in order to localize broad capabilities is an intriguing open problem.
> 3. We appreciate the reviewer's insightful idea about the Lottery Ticket Hypothesis and potential cost of applying gradient routing. We agree. We have had similar concerns backed by intuitions from Neural Tangent Kernel [1], particularly regarding the fact that in wide networks, individual parameters don't move much during training (which seems contrary to the idea of localizing to small subspaces). We do think this means that certain kinds of gradient routing schemes are "unnatural" with respect to neural net learning dynamics and will fail. So far, we have been pleasantly surprised by the success of different routing schemes, though.
>
> **Summary:**
> We understand that the reviewer finds our work sound, clearly presented, and very important, but has concerns about the strength of contribution. We have explained why we believe our contribution is significant: (a) it is a novel idea and method for localization in neural networks, and (b) it presents many new ideas for how to apply localization methods to open problems in ML. If the reviewer has further concerns we would be grateful for the opportunity to address them. Otherwise, we hope the reviewer will increase their scores to reflect the potential importance of our work.
>
> [1] Jacot, A., Gabriel, F. and Hongler, C., 2018. Neural tangent kernel: Convergence and generalization in neural networks. *Advances in neural information processing systems*, 31.

---

> ### Comment · Reviewer_Dmtv · 2024-11-22
>
> Thank you for your detailed responses. I appreciate the effort in addressing the questions, but I still have a few concerns:
>
> > I am particularly interested in the initialization process. In the MNIST generation experiments, the gradients flow through two distinct parts of the network. How are these parts initialized? Are they initialized together as in a default network, or are they initialized independently? In other words, are they initialized dependently or independently?
>
> > If Gradient Routing (GR) is used, how should the network be partitioned? Should it be divided across layers, or can different parts of a single layer also be partitioned? I would appreciate further clarification on how GR handles these scenarios.
>
> Answer these questions could help me better understand the method and help readers know how to utilize the proposed method.

---

> > ### Author Response · Authors · 2024-11-22
> >
> > > I am particularly interested in the initialization process. In the MNIST generation experiments, the gradients flow through two distinct parts of the network. How are these parts initialized? Are they initialized together as in a default network, or are they initialized independently? In other words, are they initialized dependently or independently?
> >
> > All modules are initialized using the default Pytorch initialization (as found [here](https://github.com/pytorch/pytorch/blob/8b13ed594a2b9b0a994e8efd42b8f1e59372e499/torch/nn/modules/linear.py#L114)). So, to answer your question: they were initialized together as in a default network. In our working draft, we have now clarified this point under "Training" in appendix B, which has MNIST training details. In general, we aim be explicit about all non-default choices. Along with releasing our code, we hope this makes it easy for the reader to understand and reproduce the work.
> >
> > > If Gradient Routing (GR) is used, how should the network be partitioned? Should it be divided across layers, or can different parts of a single layer also be partitioned? I would appreciate further clarification on how GR handles these scenarios.
> >
> > It depends! Gradient routing is a general method which has many special-case applications that look quite different. In appendix K (end of PDF), we discuss how to choose gradient routes from a practical perspective, and explain the choices made for the experiments in the paper. We have seen success with a variety of approaches, as described throughout the paper:
> > - In section 4.1 (MNIST), we partition the output of a single layer into two equal-sized parts corresponding to different halves of the training data. In appendix B.1, we do the same with a ResNet classifier trained on CIFAR100.
> > - In section 4.2.1 (steering scalar), we induce-axis aligned features by partitioning the outputs of adjacent intermediate layers of a Transformer into a single scalar and the remainder of the vector.
> > - In section 4.2.2-3 (unlearning), we partition several MLP layers each into two parts (the "original" part, and the smaller "expanded" part), so that the expanded part can later be ablated. We also employ negative and fractional learning rates here, so we aren't strictly partitioning the edges of the DAG.
> > - In section 4.3 (RL), we partition a policy network into two expert modules which are combined in a (learned) weighted sum.

---

> > > ### Comment · Reviewer_Dmtv · 2024-11-25
> > >
> > > Thanks for the authors responses. In response, I will raise my score to 5, as I still have a few concerns about the novelty.

---

> > > > ### Author Response · Authors · 2024-11-25
> > > >
> > > > Thank you for increasing your support for the paper. We would greatly appreciate if you could share your specific concerns about novelty so we can address them directly in the paper and ensure the contributions are presented clearly to readers. In particular,
> > > >
> > > > 1. Were there any statements in our [initial response](https://openreview.net/forum?id=z1mLNhWFyY&noteId=DhXzubTWJs) that you disagree with or are uncertain about?
> > > > 2. Are you aware of other methods that can achieve supervised control of representation learning (as in sections 4.1 and 4.2.1), or that are able to induce behavioral submodules in reinforcement learning agents despite limited labels (4.3)?
> > > >
> > > > If there are no methods that can achieve these effects, we believe our work represents a significant and novel contribution that would be of great interest to the ML community. In that case, we would respectfully ask you to raise your score to reflect this.

---

### Official Review · Reviewer_hLX2 · 2024-11-01

**Soundness:** 3
**Presentation:** 4
**Contribution:** 3
**Rating:** 6
**Confidence:** 4

**Summary:**

The paper introduces gradient routing (GR), which allows localizing computation in neural networks by masking gradients based on data-dependent paths. GR controls gradient flow within specific network subregions, isolating capacities and enabling the model to focus on particular data features / tasks. It promotes modular network design, making it suitable for selective forgetting and scalable oversight in reinforcement learning and language models. GR is evaluated on MNIST, gridworlds, and language tasks to demonstrate that it can improve task-specific performance, support targeted unlearning, and improve model interpretability by maintaining distinct functionalities in different parts of the network.

**Strengths:**

- The paper makes a new contribution to the field: GR is a new method to control gradient propagation based on data, enabling modular learning within a network.
- GR can make neural networks more interpretable by designing which part of the network is responsible for a specific task;
- GR provides support for targeted unlearning and scalable oversight;
- Broad applicability of GR (and many interesting questions to spark future work);
- GR can reduce interference and improve accuracy on specialized tasks;
- The method helps to control data access in privacy-sensitive scenarios.

**Weaknesses:**

- GR requires careful selection of mask weights, data subsets, and regions to localize. How should these be chosen? At least in its current form, the method faces difficulties scaling up to larger models.
- No comparison to similar methods mentioned that can also achieve localization (DEMix and Interchange Intervention Training, both mentioned in the text).
- Unclear relevance of GR to safety-critical applications mentioned in the paper. Demonstrating benefits in a high-stakes scenario would strengthen the paper.

**Questions:**

Q1: How the gradient masks are selected / initialized for different tasks / data samples / features? Are there meaningful heuristics that generally work well?

Q2: How to choose the network subregions that gradients should be routed through? Would certain architectures or tasks require a different approach to subregion selection? Have you experienced any achitecture-specific challenges?

---

> ### Author Response · Authors · 2024-11-19
> **Response to hLX2 (1/2)**
>
> We are very grateful to the reviewer for their thoughtful and positive review, as well as their constructive comments which have helped to improve the paper. We thank the reviewer for enumerating the strengths of the contribution, including the novelty, generality, usefulness, and interesting questions raised.
>
> The reviewer raises important points, which we address below.
>
> **Response to weaknesses:**
> > GR requires careful selection of mask weights, data subsets, and regions to localize. How should these be chosen?
>
> We have updated the paper to add a paragraph at the end of section 3, and a detailed appendix K, offering general guidance for choosing masks and explaining how we do so for our experiments.
>
> > At least in its current form, the method faces difficulties scaling up to larger models.
>
> We respectfully disagree with this statement. We have not found any evidence that gradient routing faces difficulties scaling to larger models. We show that the method scales from a 20M to a 0.7B parameter transformer without issue. (As hinted at in the limitations section, we stopped at 0.7B only due to resource limitations.) Intuitively, its simple mechanism of limiting where gradients flow seems like the kind of approach that could scale indefinitely. Of course, if a particular gradient routing application requires extensive hyperparameter tuning, that will make it cumbersome to experiment with large models in practice. This is a limitation of our current approach to robust unlearning.
>
> > No comparison to similar methods mentioned that can also achieve localization (DEMix and Interchange Intervention Training, both mentioned in the text).
>
> Thank you for highlighting this opportunity for improvement. We have added appendix J, which includes detailed comparisons to other methods and explains the unique properties of gradient routing that make it suitable to our applications. We are also in the process of re-running our experiments on TinyStories unlearning, substituting DEMix for gradient routing as a localization method. We do not plan to experiment with IIT because it requires a causal model and counterfactual versions of data points, such that it doesn't translate naturally to our applications. (However, we remain excited about IIT's potential applicability to controlling neural network internals.)
>
> > Unclear relevance of GR to safety-critical applications mentioned in the paper. Demonstrating benefits in a high-stakes scenario would strengthen the paper.
>
> We give the example of removing bioweapon-related or dual-use capabilities of AI models in section 4.2.3. We also believe that the relevance of our work to preventing Goodharting (discussed in section 4.3 and 5) is potentially a major conceptual contribution to safety in high-stakes situations, given that imperfectly-specified training objectives for a powerful AI could lead to harmful outcomes. Of course, this is a conceptual contribution only. We have also added an appendix L, in which we enumerate foundational challenges to assuring safe use of LLMs that gradient routing may help address.
>
> We strongly agree with the reviewer that the inclusion of more discussion here would strengthen the paper. Does the reviewer think that the elaboration we have added to the appendix is suitable?

---

> ### Author Response · Authors · 2024-11-19
> **Response to hLX2 (2/2)**
>
> **Response to questions:**
> > Q1: How the gradient masks are selected / initialized for different tasks / data samples / features? Are there meaningful heuristics that generally work well? Q2: How to choose the network subregions that gradients should be routed through? Would certain architectures or tasks require a different approach to subregion selection? Have you experienced any achitecture-specific challenges?
>
> We provide detailed experiment descriptions in section 4 and the appendices corresponding to each subsection (MNIST representation control: appendix B, TinyStories unlearning: appendix C, steering scalar: appendix D, virology unlearning: appendix E, and scalable oversight: appendix F). However, we thank the reviewer for raising the important practical question of how we chose these settings in the first place.
>
> Choosing a mask usually starts with a concrete idea of what capabilities are to be localized, and a motivation for doing so. We find that this starting point constraints the search space and is helpful as a guide. From that point, choosing masks is like choosing a neural net architecture: we find that a combination of intuition about neural net learning dynamics, plus trial-and-error with small-scale experiments, is effective. The newly-added appendix K gives more details on this process and may of interest.
>
> *Regarding data*: For unlearning, we were surprised to find that localizing just a few tokens that were prevalent in the forget set was sufficient to damage performance significantly (and this damage was *not* explained by bad performance on merely those tokens; see the second row of table 1 in section 4.2.3). On the other hand, attempting to localize updates for *all* tokens in the forget set led to terrible performance (high retain set training loss). This led us to the belief that for unlearning, it is important to pick a _minimal_ subset of the task that is *necessary* for good performance, and localize only that.
>
> *Regarding architectures and regions of localization*: In terms of architecture-specific challenges: we have yet to encounter an architecture that doesn't admit gradient routing. The MNIST MLP experiments worked right away and were easy to set up. The transformer experiments required more trial-and-error. In transformers, we found that localizing to MLPs was most effective (rather than attention heads or dimensions of the hidden activations). This seems consistent with evidence that MLPs serve as stores of factual information in transformer LMs [1].
>
> **Summary:**
> The reviewer's apt comments have given us an opportunity to improve the paper and clarify our findings. The paper now includes an explanation of mask selection (main body + appendix), detailed comparisons to other localization methods (appendix), and an extended discussion of gradient routing's relevance to problems in AI safety (appendix). We eagerly await other suggestions for improvements. Also, we respectfully ask the reviewer to consider raising their scores if we have addressed their concerns.
>
> [1] Kevin Meng, David Bau, Alex Andonian, and Yonatan Belinkov. Locating and editing factual associations in GPT. *Advances in Neural Information Processing Systems*, 36, 2022. arXiv:2202.05262.

---

### Official Review · Reviewer_nUgs · 2024-11-04

**Soundness:** 2
**Presentation:** 4
**Contribution:** 3
**Rating:** 5
**Confidence:** 3

**Summary:**

This paper presents a new method for training networks called Gradient Routing (GR) with the goal of isolating capabilities to specific subregions of the network by controlling which network subregions are updated by which data points.

The paper shows applications for gradient routing towards a variety of settings and problems:

- The authors first show that by applying GR to an MNIST-autoencoder, it is possible to isolate the representation of digits 0-4 to the first half of an embedding and digits 5-9 to the second half. Additionally, the authors show that GR can be used for activation steering in language models.

- The authors then propose to use GR for robust unlearning in already learned language models, via an approach called expand-route-ablate (ERA), where pretrained networks are expanded to include new subregions, some already learned capabilities are re-routed to new subregions, and then new subregions are ablated from the model, unlearning these previously learned capabilities from the network. Experiments show that ERA is successful in unlearning topics for small language models, even if only a fraction of "to-unlearn" samples are labelled.

- The paper then shows that vanilla GR can be successfully applied to learn a 0.7B language model that lacks specific harmful capabilities such as bioweapon related capabilities.

- Finally, the paper demonstrates an application of GR in reinforcement learning, to learn a policy that avoids certain target squares in a grid using a limited oversight signal to route gradients.

**Strengths:**

The proposed method of gradient routing is novel and interesting, and has applications for multiple important problems in safety and interpretability.

Impressively, the proposed method enables forgetting undesirably capabilities even without labelling full forget sets.

**Weaknesses:**

- The vanilla GR method requires splitting data into "retain" and "forget" sets before training begins, and then routing gradients during a potentially long and expensive training process. This limits applicability (except of the ERA method) and raises the question if GR is ever necessary for language model unlearning applications. If we knew the split during pretraining, why would we not simply omit the forget set during pretraining, learning the "pure" model considered in the paper?

- The ERA method requires more experiments and analysis/ablations to show why it works -- it is not clear why it is possible to introduce new network units, set low learning rate on old units for forget set samples, ablate the new units, and observe poor performance on forget set samples when running the old units. Is this a one-off result? Does this generalize to more tests and bigger models that may have more significant understanding of forget-set content?

- GR requires isolating each capability to a subset of examples before training and establishing network subregions for each capability, which may be challenging when there are lots of examples or lots of capabilities.

**Questions:**

Why does "absorption" happen? Is the absorption effect reliable across multiple settings?

Questions on the robust unlearning experiment
- In Fig 4(a), why does the validation loss increase with more training steps? Can we see what happens with a more realistic number of finetuning steps (say 1000)?
- Why does RMU appear to outperform ERA on the 4-stories task? Does it just have overall higher loss?
- In fig 4(b) it appears that post-finetuning retain set los is also increased. How much of the validation forget loss in ERA and RMU is just due to overall slightly worse models?

Is ERA expected to work when the pretraining model and dataset are big (such as for modern LLMs) and the representations in the original model have good understanding of the forget set?

---

> ### Author Response · Authors · 2024-11-19
> **Response to nUgs (1/2)**
>
> We thank the reviewer for their thoughtful engagement with the entirety of the paper. We are glad that they found gradient routing "novel and interesting" and relevant to "multiple important problems in safety and interpretability." We are also happy that they acknowledge gradient routing's usefulness in the absence of perfect data labeling. These are the reasons we are excited about gradient routing, too.
>
> The reviewer rightfully identifies limitations with our study of robust unlearning, which we address below.
>
> **Response to weaknesses:**
> > The vanilla GR method requires splitting data into "retain" and "forget" sets before training begins, and then routing gradients during a potentially long and expensive training process. This limits applicability (except of the ERA method) and raises the question if GR is ever necessary for language model unlearning applications.
>
> This is correct: conventional post-hoc unlearning methods do not require data labeling during pre-training. However, as explained in section 2 (and the top-level comment), conventional unlearning methods are "shallow", acting merely as "safety wrappers" rather than robustly removing harmful capabilities. In this case, our approach sacrifices generality and ease of use in exchange for *actually removing the capabilities*. As mentioned in section 4.2.3, one such application of this method would be the removal of high-stakes information, like bioweapon-related capabilities.
>
> > If we knew the split during pretraining, why would we not simply omit the forget set during pretraining, learning the "pure" model considered in the paper?
>
> There are many settings where data filtering is ineffective  or infeasible, either because of _imperfect labeling_ (discussed throughout the paper, including section 4.3 and the first part of the discussion section; the effect is also quantified at the end of section 4.2 and in figure 10 in appendix C.1.) or because of _entangled capabilities_ (when some capabilities imply or are implied by other capabilities, discussed in second part of section 5).
>
> > The ERA method requires more experiments and analysis/ablations to show why it works -- it is not clear why it is possible to introduce new network units, set low learning rate on old units for forget set samples, ablate the new units, and observe poor performance on forget set samples when running the old units. Is this a one-off result? Does this generalize to more tests and bigger models that may have more significant understanding of forget-set content?
>
> We present ERA as a pre-training technique: the model is trained from scratch, with the forget and retain gradients being routed to different parts of the model from the beginning. This method should work in general because we are constraining the learning updates for forget data to occur in parameters which we will then delete.
>
> In sections 4.2.2. and 4.2.3, we see that the method works when applied across two model sizes (small 28M, large 0.7B), and two datasets (TinyStories, FineWeb-Edu / WMDP). In section 4.2.2., we provide extensive comparisons with a variety of baselines, including a gold-standard *pure* model trained on retain data only, a *control* model, which is identical to ERA but does not apply gradient routing, etc. These comparisons show that routing is responsible for localization and provide comprehensive metrics.
>
> > GR requires isolating each capability to a subset of examples before training and establishing network subregions for each capability, which may be challenging when there are lots of examples or lots of capabilities.
>
> This is a good point and a genuine challenge to potential gradient routing applications. As mentioned in the introduction and section 3, we imagine gradient routing being used to localize a limited set of capabilities in order to provide targeted guarantees about model behavior, for safety purposes. We share the reviewer's perspective that localizing many capabilities will be challenging. In particular, it would likely harm model performance too much to be useful.
>
> That said, we are excited by the results on absorption, which suggest that exhaustive examples may not be required to localize capabilities (see, e.g., the results of applying gradient routing to only the `_California` token in section 4.2.1). Instead, a set of limited examples may be sufficient.

---

> ### Author Response · Authors · 2024-11-19
> **Response to nUgs (2/2)**
>
> **Response to questions:**
> > Why does "absorption" happen? Is the absorption effect reliable across multiple settings?
>
> As mentioned in the first paragraph of the discussion section (section 5), we see evidence for absorption in multiple settings: "... an i.i.d. subset of the data (TinyStories unlearning in section 4.2.2), and for semantically limited data (steering scalar in section 4.2.1, virology unlearning in section 4.2.3, scalable oversight in section 4.3)." After, we hypothesize about why this happens.
>
> Beyond the explanation given in the paper, there is a simple dynamic that likely explains part of the absorption effect: if a feature exists (in a hidden layer) that is predictive for the downstream task, then it is "easier" for the neural net to learn to utilize this feature than to re-learn the feature in another location. In other words, there may a general inductive bias in neural nets away from feature duplication. If so, inducing a feature to be learned in one location makes it less likely to be learned in another. This is especially true when explicit regularization is applied.
>
> On robust unlearning:
> > In Fig 4(a), why does the validation loss increase with more training steps? Can we see what happens with a more realistic number of finetuning steps (say 1000)?
>
> The validation loss increases because the model overfits to the small number of training points. It will continue to overfit if trained longer. Our approach of training on a few samples is standard in the literature: we cite a variety of works that do this in the discussion of robust unlearning in section 2, for example, [1] and [2]. This is a natural thing to do because, as it turns out, conventional unlearning methods can be undone by fine-tuning on _very_ small samples of forget sets. These sets can be as small as **2** (see [1], table 5)! In contrast, our experiments show that ERA is approximately as robust to retraining as a gold-standard "pure" model that was never trained on forget data. This is evidence that we have truly removed capabilities from the model.
>
> > Why does RMU appear to outperform ERA on the 4-stories task?
>
> Like many unlearning methods, RMU works by directly degrading the model's performance on the forget set. In contrast, ERA doesn't target high forget loss directly. Instead, high forget loss for ERA is a side effect of ablating network components responsible for forget set performance. This difference is reflected in the forget set retrainability of RMU vs. ERA, showing that the latter is much harder to retrain (on par with a gold-standard "pure" model never trained on forget data). However, it happens that in this case, fine-tuning with only 4 samples isn't enough to overcome RMU's degraded forget set performance. (16 is sufficient.)
>
> > In fig 4(b) it appears that post-finetuning retain set loss is also increased. How much of the validation forget loss in ERA and RMU is just due to overall slightly worse models?
>
> This is a great question! We were concerned about this as well. We quantify this effect by comparison to two other models: a "base" model, trained conventionally (see "Alignment tax" in 4.2.2), and a "control" model (trained identically to ERA, but without gradient routing) (see Figure 4b). The answer is that some of the effect (0.1 cross entropy) is due to the model simply being worse at fitting the data, but the majority of the effect comes from ablation (0.4 cross entropy).
>
> > Is ERA expected to work when the pretraining model and dataset are big (such as for modern LLMs) and the representations in the original model have good understanding of the forget set?
>
> We think this is answered by our clarification that ERA is a pre-training method: ERA works by controlling how representations form in the first place, not by modifying existing representations. It is thus applicable to any scale of LLM.
>
> **Summary:**
>
> We are grateful for the reviewer's attention to detail in our unlearning results. We hope we have satisfactorily demonstrated the soundness of the work by clarifying the interpretation of results and the extent of our experiments. If the reviewer has any suggestions for improving the clarity here (while respecting the page limit), we would be grateful. We also ask that the reviewer consider increasing their score if they feel we have addressed their points adequately.
>
> [1]: Sheshadri, A., Ewart, A., Guo, P., Lynch, A., Wu, C., Hebbar, V., ... & Casper, S. (2024). Latent Adversarial Training Improves Robustness to Persistent Harmful Behaviors in LLMs. arXiv preprint arXiv:2407.15549.
>
> [2]: Deeb, A., & Roger, F. (2024). Do Unlearning Methods Remove Information from Language Model Weights?. arXiv preprint arXiv:2410.08827.

---

> ### Comment · Reviewer_nUgs · 2024-11-25
> **Response to authors**
>
> Thanks to the authors for their thoughtful answers to all my questions, and for clarifying that ERA is a pre-training technique and not one that can be applied to already trained models. I have two major remaining concerns that prevent me from raising my score.
>
> My first concern is that the value of the contribution may be very limited due to the proposed method only being applicable in contrived settings: where we know what the undesirable capabilities are before training, and yet have only imperfect labeling (or "entangled capabilities") of data that correspond to those capabilities. The paper needs more motivation for why studying this setting is important.
>
> My second (and bigger) concern is that the experimental results do not appear to suggest that gradient routing meaningfully outperforms the baseline of data filtering in these settings (RMU is unfair to use as a baseline since it can be applied to models that have already been trained). Thanks to the authors for highlighting the experiment in Appendix C comparing robust unlearning performance of ERA to data filtering when only limited filtering is possible. While Figs 10 and 11 show that ERA does outperform filtering when a low proportion of forget stories are labeled, the performance difference appears to be extremely marginal, and presumably the data filtering model has lower overall validation loss that may explain even that difference (and lower implementation complexity - no need for potentially expensive modifications to the pre-training method). As a result, I am unconvinced of the utility of gradient routing for the robust unlearning problem from the experimental results in the paper. I urge the authors to find an experimental setting where the data has imperfect labeling or "entangled capabilities" and show that gradient routing can actually meaningfully outperform data filtering in that setting.

---

> > ### Author Response · Authors · 2024-11-29
> >
> > We thank the reviewer for their engagement and thoughtful feedback. It has been quite helpful. We have uploaded a significant revision (changelog **[here](https://openreview.net/forum?id=z1mLNhWFyY&noteId=upQaj3O7So)**) to address the remaining concerns. We also comment here.
> >
> > > My first concern is that the value of the contribution may be very limited due to the proposed method only being applicable in contrived settings: where we know what the undesirable capabilities are before training, and yet have only imperfect labeling (or "entangled capabilities") of data that correspond to those capabilities. The paper needs more motivation for why studying this setting is important.
> >
> > We respectfully disagree that our setting is contrived.
> >
> > * In real-world ML applications, label availability and quality is often a bottleneck. This is the motivation for semi-supervised learning, transfer learning, and scalable oversight, for example. Also,  practical challenges to data filtering are well-documented. We have updated the introduction, related works, first paragraph of section 4.3, and the discussion to reflect this.
> > * There are known undesired capabilities that are thoroughly tested in the release of new frontier language models. [Anthropic](https://www-cdn.anthropic.com/fed9cc193a14b84131812372d8d5857f8f304c52/Model_Card_Claude_3_Addendum.pdf) and [OpenAI](https://openai.com/index/openai-o1-system-card/?ref=planned-obsolescence.org) test models for risky capabilities including Cybersecurity, [CBRN](https://en.wikipedia.org/wiki/CBRN_defense) and persuasion capabilities. In section 4.2.3, we motivated the removal of virology capabilities by dual use concerns [1, 2].
> > * We discuss related ideas in the newly-improved discussion section and appendix L: _Relevance of gradient routing to problems in AI safety_.
> >
> > > My second (and bigger) concern is that the experimental results do not appear to suggest that gradient routing meaningfully outperforms the baseline of data filtering in these settings (RMU is unfair to use as a baseline since it can be applied to models that have already been trained).
> >
> > This is a great point. The metrics we previously reported did not highlight the most important comparisons. We've completely revised the presentation of our results to highlight key comparisons. This establishes ERA's superior performance in the partial labeling regime. We also increase sample sizes and compare against an alternative localization-based method, DEMix (plus ablation).
> >
> > We feel that the comparison to RMU (as a stand-in for conventional post-hoc unlearning methods) is scientifically meaningful, as it contrasts the "deepness" of ERA unlearning with the "shallowness" of post-hoc unlearning methods in general (see "Robust unlearning" in the related works section). However, we agree with the reviewer that it would not be fair to conclude that gradient routing is a strict improvement on RMU.
> >
> > [1] Fabio Urbina et al, 2022. [Dual use of artificial-intelligence powered drug discovery](https://www.nature.com/articles/s42256-022-00465-9). Nature Machine Intelligence, 4(3):189–191. ISSN 2522-
> > 5839. doi: 10.1038/s42256-022-00465-9.
> >
> > [2] Nathaniel Li et al, 2024. [The WMDP Benchmark: Measuring and Reducing Malicious Use with Unlearning](https://proceedings.mlr.press/v235/li24bc.html). Proceedings of the 41st International Conference on Machine Learning, PMLR 235:28525-28550.

---

### Author Response · Authors · 2024-11-19
**Thank you to all reviewers, plus clarification of contributions**

We thank the reviewers for their thoughtful comments and astute questions, which have already helped us make significant improvements to the paper. In addition to revisions to the paper, we include a general response here, as well as detailed responses to each reviewer below.

## Clarifying our scholarly contributions
We respectfully submit that the reviewers have  underestimated the strength of the contributions of the paper, which are as follows:
1. **We establish a novel way to control neural network internals that has unique, desirable properties**: Gradient routing allows the user to flexibly assign learning updates to network regions at any level of granularity and in any training regime. To the best of our knowledge, no other method exists that is relevant to all of the applications areas we present. We have added a new section, appendix J: Extended Comparisons to Other Modularity Methods, to clarify these claims and provide detailed comparisons to existing methods.
3. **Using gradient routing, we demonstrate progress on three major open problems in machine learning**:
    * supervised control of feature learning, an open problem not solved by the subfield of adversarial representation learning (mentioned in section 2 of the paper), nor by disentangled representation learning (an unsupervised approach mentioned in appendix I);
    * robust removal of harmful capabilities, an open problem that has received substantial attention due to a recent flurry of works establishing the "shallowness" of typical unlearning methods, as we discuss in section 2; and
    * scalable oversight, a general open challenge to supervised training of advanced AI systems (discussed in sections 2, 4.3, 5, and at length in the appendix).
4. Our work additionally **contains insights and raises intriguing questions about neural net learning dynamics and AI control**.
    * We report results from a variety of experiments suggesting the existence of an *absorption* effect of feature localization (as discussed in section 5).
    * To the best of our knowledge, we are also the first to consider the problem of unlearning that is robust to *imperfect labeling*.
    * We identify a general strategy for solving the problem of *Goodharting* [1][2], by utilizing imperfect labels to influence behavior without including those labels in the behavioral training objective (as demonstrated in section 4.3, discussed in section 5, and elaborated on in the appendix).
    * We introduce a simple-yet-challenging toy model for scalable oversight by noting a key challenge to achieving weak-to-strong generalization [3]: that a policy can condition on problem difficulty.

Finally, we urge the reviewers to consider the breadth of our investigations: we have applied gradient routing to feedfoward MLP autoencoders, deep CNN classifiers (newly added per reviewer suggestion), transformer language models of varying sizes, and reinforcement learning agents. We show that gradient routing is applicable across these diverse domains.

We believe that the above represents a significant contribution to the literature. If we are mistaken, we would be grateful for any corrections or feedback reviewers can offer.

## Putting gradient routing in context
Gradient routing is not intended to be competitive with conventional supervised learning methods (which target low loss, without concern for internal mechanisms), nor with conventional post-hoc unlearning methods (which target high forget-set loss, **not** robust removal of capabilities). The purpose of gradient routing is to enable new approaches to tackling foundational challenges presented by the development of powerful AI systems [4]. We reference some of these challenges in the introduction and discussion sections. To clarify the contribution, we have added further discussion in appendix L.

[1] C. A. E. Goodhart, 1984, _Problems of Monetary Management: The UK Experience_, Macmillan Education UK, London, ISBN 978-1-349-17295-5.

[2] Karwowski et al., 2024, _Goodhart's Law in Reinforcement Learning_, The Twelfth International Conference on Learning Representations, [openreview.net/forum?id=5o9G4XF1LI](https://openreview.net/forum?id=5o9G4XF1LI).

[3] Burns et al., 2024. Weak-to-Strong Generalization: Eliciting Strong Capabilities With Weak Supervision. _Proceedings of the 41st International Conference on Machine Learning_, 235:4971-5012.

[4] Anwar et al., 2024, _Foundational Challenges in Assuring Alignment and Safety of Large Language Models_, Transactions on Machine Learning Research, 2835-8856, 2024.

---

> ### Comment · Area_Chair_AYCA · 2024-11-19
> **Discussion**
>
> Dear Reviewers,
>
> Thanks for the reviews. The authors have uploaded their responses to your comments, please check if the rebuttal address your concerns and if you have further questions/comments to discuss with the authors. If the authors have addressed your concerns, please adjust your rating accordingly.
>
> AC

---

### Author Response · Authors · 2024-11-28
**A significant final revision**

We have uploaded a significant revision in response to reviewer feedback. We are indebted to the reviewers whose engagement with the work and thoughtful suggestions enabled these changes.

## Changelog

* **Introduction**
  * Improved the writing to more clearly explain our conceptual and practical contributions.
* **Background and related work**
  *  Added a review of the limitations of data filtering for control of neural network behavior, improving the motivation for gradient routing.
* **TinyStories unlearning (section 4.2.2)**
   * Added _DEMiX plus ablation_ so that ERA can be compared to another localization-based method. Contrasting the two approaches provides strong evidence of _absorption_ (see section 5). (We thank Reviewer hLX2 for the suggestion.)
   * Changed our metrics to highlight the most important comparisons: forget set performance before/after unlearning and retain set performance. This controls for the possibility that a method is simply worse at fitting the data. (We thank Reviewer nUgs for the suggestion.)
   * We report strong results clearly establishing ERA's superior performance in the partial labeling regime.
   * We add a result on ERA plus RMU that outperforms either method alone at robust unlearning. (This combination is natural because RMU is a "light touch" post-hoc method that does negligible damage to retain set performance.)
* **Scalable oversight in reinforcement learning (section 4.3)** We noticed that reviewers have not focused much on this part of the paper, which we feel is the most important part, conceptually.
   * To highlight the contribution, we re-wrote the section, adding clearer motivation and presentation.
   * We report better results based on improvements to our implementation: gradient routing induces specialized submodules even at 1% data labeling and achieves massive gains in data efficiency as compared with baselines.
   * We re-wrote appendix F and added learning curves and ablations to provide better context to the results in the main body of the text.
* **Discussion (section 5)**
   * Expanded discussion of absorption by contrasting ERA to DEMix and improved our explanation of the effect.
   * Added subsections "Mechanistic supervision avoids Goodharting" and "Entangled capabilities motivate gradient routing," further clarifying the strong conceptual motivation for gradient routing.
   * Added further motivation for gradient routing by proposing a concrete way that localization could enable traditional computer security approaches [1] to high-stakes AI deployment. (Discussed further in our newly-improved appendix L, on gradient routing's applicability to problems in AI safety.)


[1] R. S. Sandhu and P. Samarati, "Access control: principle and practice," in IEEE Communications Magazine, vol. 32, no. 9, pp. 40-48, Sept. 1994, doi: 10.1109/35.312842.

---

### Meta-Review · Area_Chair_AYCA · 2024-12-18

**Metareview:**

(a) summary

This paper addresses some limitations with the current neural network training algorithms relating to safety. It proposes to apply data-dependent masks provided by users for routing backpropagated gradients to the specified subregions in a neural network. Constraining weight updating to isolated subregions help learn partitioned representation, and is suitable for selective forgetting and scalable oversight reinforcement learning.

(b) strengths
+ The gradient routing is useful for deploying AI systems in high-risk scenarios.
+ It is applicable to broad cases including unlearn tasks, protecting privacy.
+ It improves accuracy with reduced interference.

(c) weaknesses
- The proposed method requires careful selection of hyper-parameters which limits its application.
- Gradient routing does not outperforms the baseline of data filtering in the experimental settings.
- There is an issue with scalability when there are lots of examples or lots of capabilities.
- It has limited novelty: similar ideas have already been proposed by previous papers (1) Piggyback [1] and PackNet[2] tries to learn different tasks better with different subsets of the weights by leveraging a binary mask. (2) Parameter efficient tuning method for Large language models(LLMs) such as lora[3], can efficiently learn different downstream tasks using different adaptor modules.

(d) decision

The proposed gradient routing method is interesting and has potential application to broad use cases. However, the issues with sensitivity to hyper-parameters selection and scalability affects its usefulness. The limited novelty also makes it not  ready for publication at ICLR. Please keep the reviewer comments in mind when preparing a future version of the manuscript.

**Additional Comments On Reviewer Discussion:**

The reviewers think the paper is well-motivated and the proposed method is applicable to broad cases, however they also share the concerns on the usefulness and scalability of the proposed method due to sensitivity of hyper-parameters selection, and challenging when there are lots of examples or lots of capabilities. Another concern is the limited novelty -- similar ideas have been proposed in previous work. The authors' rebuttal and updated manuscript addressed some concerns, however,
 the reviewers are not convinced of the utility of gradient routing from the experimental results in the paper.

---

> ### Public Comment · ~Alex_Cloud1 · 2025-02-06
> **Clarifying the findings of the paper**
>
> Hello to future readers!
>
> Thank you for your interest in gradient routing. To prevent possible confusion arising from the review, I clarify a few points here.
>
> **Gradient routing is not always sensitive to hyperparameters.** We did *not* observe noteworthy sensitivity to hyperparameters in our experiments on MNIST, CIFAR, or reinforcement learning.
>
> **Gradient routing outperforms data filtering.** A key finding of the paper is that gradient routing outperforms data filtering when data is partially labeled. This held for the two settings we tested: language model unlearning (section 4.2.2) and scalable oversight of reinforcement learning (section 4.3). These results are discussed in the introduction and discussion sections as one of the primary motivations for the work.
>
> **We [suspect](https://openreview.net/forum?id=z1mLNhWFyY&noteId=UNhSuUxKel) that gradient routing will scale gracefully to larger models and datasets**, although we can't be certain without further experiments. In the appendix, we state that "we suspect that naive attempts to localize large numbers of concepts to unique regions will lead to high training loss." This is not a limitation of gradient routing. Rather, it is a hypothesis about a highly "unnatural" learning target. Put differently, we expect that *any* method that tries to modularize large numbers of capabilities to distinct subregions will incur high training loss. Of course, such a feat is not necessary for a method to be impactful. Our robust unlearning application requires localizing merely a single data source (the forget set) to a single location.
>
> **Gradient routing and its applications are novel,** as explained [here](https://openreview.net/forum?id=z1mLNhWFyY&noteId=DhXzubTWJs), in extensive comparisons to other methods in [Appendix J](https://openreview.net/pdf?id=z1mLNhWFyY#page=41), and in the clarification of our contributions [here](https://openreview.net/forum?id=z1mLNhWFyY&noteId=NhpNRz9eSb). Gradient routing _generalizes_ previous methods, and furthermore, our safety- and control-oriented applications are, to the best of our knowledge, previously unimagined.

---

### Decision · Program_Chairs · 2025-01-22

Reject